# Generating Teammates for Training Robust Ad Hoc Teamwork Agents via Best-Response Diversity

**Arrasy Rahman**                                                    *arrasy@cs.utexas.edu*
*Department of Computer Science*
*University of Texas at Austin*

**Elliot Fosong**                                                    *e.fosong@ed.ac.uk*
*School of Informatics*
*University of Edinburgh*

**Ignacio Carlucho**                                                *ignacio.carlucho@ed.ac.uk*
*School of Informatics*
*University of Edinburgh*

**Stefano V. Albrecht**                                            *s.albrecht@ed.ac.uk*
*School of Informatics*
*University of Edinburgh*

**Reviewed on OpenReview:** *https://openreview.net/forum?id=l5BzfQhROl*

## Abstract

Ad hoc teamwork (AHT) is the challenge of designing a robust learner agent that effectively collaborates with unknown teammates without prior coordination mechanisms. Early approaches address the AHT challenge by training the learner with a diverse set of handcrafted teammate policies, usually designed based on an expert's domain knowledge about the policies the learner may encounter. However, implementing teammate policies for training based on domain knowledge is not always feasible. In such cases, recent approaches attempted to improve the robustness of the learner by training it with teammate policies generated by optimising information-theoretic diversity metrics. The problem with optimising existing information-theoretic diversity metrics for teammate policy generation is the emergence of superficially different teammates. When used for AHT training, superficially different teammate behaviours may not improve a learner's robustness during collaboration with unknown teammates. In this paper, we present an automated teammate policy generation method optimising the Best-Response Diversity (BRDiv) metric, which measures diversity based on the compatibility of teammate policies in terms of returns. We evaluate our approach in environments with multiple valid coordination strategies, comparing against methods optimising information-theoretic diversity metrics and an ablation not optimising any diversity metric. Our experiments indicate that optimising BRDiv yields a diverse set of training teammate policies that improve the learner's performance relative to previous teammate generation approaches when collaborating with near-optimal previously unseen teammate policies.

## 1 Introduction

Ad hoc teamwork (AHT) is the challenging problem of creating a single agent, called the *learner*, which can robustly collaborate with a set of unknown teammates without prior coordination mechanisms (Stone et al., 2010; Mirsky et al., 2022). Although teammates in AHT are assumed to be working together to achieve a common goal, they may exhibit different behaviours or assume different roles in the team. Attaining optimal

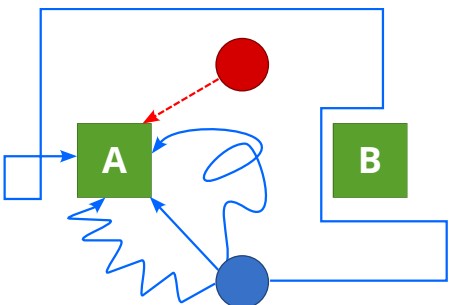

(a) Superficially different generated team-mate policies with high trajectory diversity.

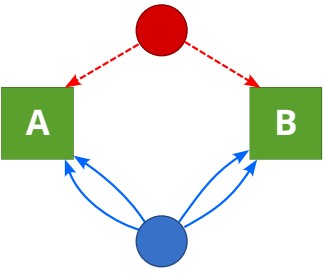

(b) Generated teammate policies by optimizing best-response diversity.

Figure 1: **Improving learner robustness by reducing superficial differences between generated teammates.** Figures 1a and 1b show an illustrative example of training an AHT learner with different sets of generated teammate policies. In this example, the learner (red dot) and its teammate (blue dot) must move to the same landmark (green rectangles) to get rewarded. Following the larger variation within the trajectory generated by different policies (blue arrow), Figure 1a shows teammate policies with higher trajectory diversity. At the same time, the illustrated policies also contain high superficial differences since a common best-response policy can effectively deal with each policy. Training a reward-maximising learner against teammates from Figure 1a will result in a learner that only moves towards landmark A, which will cause the learner to produce highly suboptimal returns when dealing with teammates moving towards landmark B. Meanwhile, Figure 1b shows teammate policies with fewer superficial differences, whose trajectories require two best-response policies corresponding to the movement towards landmarks A and B. An AHT learner trained with these teammates will learn to follow their teammate to any landmark. This is symbolised by the red arrows pointing to landmark A and B. Figure 1a and 1b illustrate how reducing superficial differences between teammate policies intended for AHT training can improve the learner robustness, indicated by the learner's ability to achieve high returns when collaborating with a broader range of teammates.

collaboration with teammates with different policies and roles may require the learner to use distinct policies. To solve AHT's challenge of optimal collaboration with unknown teammates, a robust AHT learner must then adapt its policy based on the teammates' displayed behaviour and the current team composition.

Prior AHT approaches train a learner by allowing it to interact with different teammates during training. These interaction experiences are utilised to approximate the best-response policy to each teammate by means of reinforcement learning. The best-response policy to each encountered teammate enables the learner to maximise its returns when collaborating with that particular teammate. Alongside the best-response policies, AHT approaches identify unique characteristics that differentiate the teammates' behaviour. The learner then decides its action at evaluation time by initially inferring whether the unknown teammates' behaviour displays specific characteristics identified during training. Based on their identified characteristics, a learner approximates the optimal policy for collaborating with unknown teammates by generalising the best-response policies learned for teammates encountered in training. Existing AHT approaches often use policy mixtures (Albrecht et al., 2016; Barrett et al., 2017) or neural networks (Rahman et al., 2021; Papoudakis et al., 2021a; Zintgraf et al., 2021) as generalisation models that extrapolate the best-response policies designed for training teammates towards new teammates with unknown policies.

Following its importance for AHT training, designing a diverse collection of teammate policies is crucial for training a robust learner. Careful design of such a collection is especially required in problems where there are different possible teammate policies which require different best-response policies. Failure to identify the teammate policies requiring different best-response policies for AHT training can cause the learner to be less robust. A learner's lack of robustness stems from existing AHT approaches only learning the best-response policy to teammates encountered during training. The learner's returns are more likely to be suboptimal when paired with teammates whose best-response policies are not learned during training. Although its generalisation model alleviates this problem, training a learner to collaborate with all teammate policies requiring different best-response policies remains a more reliable way to improve its robustness.

Previous approaches for designing teammate policies for AHT training fall into two categories. First, early AHT approaches (Albrecht & Ramamoorthy, 2013; Barrett et al., 2014; Albrecht et al., 2016; Barrett et al., 2017) formulate training teammate policies based on experts' knowledge regarding reasonable teammate behaviours an agent may encounter in an environment. Second, more recent AHT approaches (Xing et al., 2021; Lupu et al., 2021; Lucas & Allen, 2022) generate diverse teammate policies by optimising information-theoretic diversity metrics, which encourage an increased difference of the trajectory or action distribution between generated teammate policies.

In terms of facilitating the emergence of robust learners through AHT training, existing teammate policy generation methods face significant problems. Methods that rely on an expert's domain knowledge to formulate reasonable teammate policies have limited applicability since such knowledge is often unavailable or difficult to elicit in many real-world AHT problems. Meanwhile, teammate generation methods that maximise information-theoretic metrics may produce teammates with distinct trajectory distributions despite having the same best-response policy (Lupu et al., 2021). We refer to differences between these teammate policies that the same best-response policy can optimally deal with as *superficial differences*. Teammates with superficial differences are redundant for improving robustness because training with them will encourage AHT learners to learn the same optimal policy. In a typical teammate generation process where only a finite number of teammates can be generated, such redundancy should be reduced to encourage learners to become more robust by instead learning a broader range of best-response policies during training. We further illustrate this need to reduce superficial differences between generated teammate policies in Figure 1.

In this work, we present a teammate generation method which discourages the emergence of teammate policies with superficial differences by optimising a diversity metric called **B**est-**R**esponse **Div**ersity (BRDiv)[1]. Instead of assessing diversity in terms of information-theoretic measures, BRDiv measures diversity based on returns. To compute diversity, we measure the returns obtained from pairing any policy in the set of generated policies ($\Pi^{\text{train}}$) with policies from the set of best-responses to $\Pi^{\text{train}}$, defined as $\text{BR}(\Pi^{\text{train}})$. We empirically demonstrate that BRDiv prevents the emergence of teammate policies with superficial differences in their behaviour. This improvement is achieved by optimising the best-response policy for a teammate policy to produce low returns when collaborating with other teammate policies. The BRDiv metric can be optimised using off-the-shelf MARL techniques to produce teammate policies with minimal superficial differences. Our experiments compare the returns of a learner trained with teammate policies generated by BRDiv, previous teammate generation approaches based on action and trajectory diversity maximisation (Lupu et al., 2021; Lucas & Allen, 2022), and an ablation of BRDiv that trains teammates' policies independently. We empirically demonstrate the robustness of a learner trained with teammate policies generated by BRDiv, by showing that it achieves higher returns than other evaluated baselines when paired against near-optimal previously unseen teammate policies.

## 2 Related Work

**Ad Hoc Teamwork (AHT).** AHT was defined as a formal challenge of developing a learner capable of collaborating with unknown teammates by Stone et al. (2010). Since then, previous works (Mirsky et al., 2022) have explored AHT under different application areas and alternative names, such as zero-shot coordination (ZSC) (Hu et al., 2020) which explores AHT in problems where unknown teammates are optimal agents optimising the same reward function as the learner. Many of these works utilise type-based methods (Albrecht et al., 2016; Barrett et al., 2017; Albrecht & Stone, 2018; Rahman et al., 2021; 2022). A limitation of type-based approaches is that they assume access to predefined teammate policies for learning. This entails the manual specification of all possible types, which is often an infeasible process. Our work seeks to bridge this gap by providing ways to automatically generate teammates.

**Multi-agent Reinforcement Learning (MARL).** The objective of MARL is to jointly train a set of agents to maximise their returns in the presence of each other (Papoudakis et al., 2021b; Zhang et al., 2021). Unlike ad hoc teamwork, these methods assume full control of all members of the team. Current methods in the literature have shown great success in solving complex tasks (Vinyals et al., 2019; Christianos et al., 2021), and have been shown to be able to adapt to novel tasks (Schäfer et al., 2022). However, a drawback of

---

[1]Implementation code is provided here: `https://github.com/uoe-agents/BRDiv`

joint training is that the resulting agents have low performance when interacting with agents that are not encountered during the joint training process (Vezhnevets et al., 2020; Hu et al., 2020; Rahman et al., 2021).

**Teammate Policy Generation.** Diverse teammate policy generation has been previously explored in problems that are closely related to AHT, such as in zero-shot coordination (ZSC) (Hu et al., 2020). Several works in this area formulate diversity in terms of information-theoretic measures defined over the generated policies' trajectories (Xing et al., 2021; Lupu et al., 2021; Lucas & Allen, 2022). Despite its prevalence, previous works (Lupu et al., 2021; Liu et al., 2021) highlighted that training with teammates generated by trajectory diversity-based methods does not always lead to improved learner's robustness, which we also demonstrate through our experiments. This is because many teammate behaviours producing distinct trajectories entail the same learner's best-response policy. While Liu et al. (2021) also proposed an approach based on the best-response policies' performance, their approach is limited to zero-sum games.

**Diversity in Reinforcement Learning.** In single-agent reinforcement learning, diversity maximisation is mainly utilised as a way for agents to increase exploration (Pathak et al., 2017; Hong et al., 2018; Parker-Holder et al., 2020) or discover reusable skills (Eysenbach et al., 2019). For example, Eysenbach et al. (2019) proposed a method to learn a diverse set of reusable skills by only maximising an information-theoretic objective. Similarly, in multi-agent reinforcement learning (MARL), works such as MAVEN (Mahajan et al., 2019), have aided exploration by maximising a mutual-information metric between the trajectories and a latent space. Another recent work also utilised reward randomisation to achieve diverse behaviours in multi-agent settings (Tang et al., 2021). As another application of diversity optimisation in RL, Li et al. (2021) proposed a method optimising an information-theoretic objective to facilitate agents' specialisation towards a diverse range of roles for solving a MARL problem. Note that unlike when inducing diversity for teammate policy generation, these techniques are not designed to create a diverse set of teammates to improve the robustness of a learner.

**Population-based Training (PBT).** Our method aims to train a population of agent policies that optimise a specific metric, similar to existing works on population-based training. Population based training was proposed by Jaderberg et al. (2017) as a way to speed up the optimisation process of neural networks. This asynchronous algorithm jointly optimises a population of models and their respective hyperparameters, through an alternating process of parallel training and hyperparameter tuning. Further work from Li et al. (2019) then introduced a framework that enables population-based training in more general settings. Unlike our method which optimises the diversity of the entire population, note that PBT methods optimise an objective function defined over a single individual. PBT then uses its population of agents to iteratively generate new population members having more optimal objective function values, which is different from our method's use of MARL techniques for optimisation.

## 3 Background and Setting

In this section, we formalises the problem of teammate policy generation. We first start by formalising the interaction between agents in our AHT problem. We then provide details on the main objective of a teammate generation process given our previous formulation of agents' interaction.

### 3.1 Decentralised Partially Observable Markov Decision Process

We model the interaction between agents in a AHT environment as a decentralised partially observable Markov decision process (Dec-POMDP) (Bernstein et al., 2002). Dec-POMDPs are formally defined as an 8-tuple, $\langle N, S, \{\mathcal{A}^i\}_{i=1}^{|N|}, P, R, \{\Omega^i\}_{i=1}^{|N|}, O, \gamma \rangle$. Within a Dec-POMDP, $N$, $\mathcal{S}$, and $\gamma$ denote the set of agents, state space, and discount rate, respectively. $\mathcal{A}^i$ and $\Omega^i$ represent the action and observation space of agent $i$, respectively. The transition function of a Dec-POMDP is denoted by $P : S \times \mathcal{A}^1 \times \cdots \times \mathcal{A}^{|N|} \mapsto \Delta S$, where $\Delta S$ represents the set of all possible probability distributions over $S$. Similarly, the reward function is denoted by $R : S \times \mathcal{A}^1 \times \cdots \times \mathcal{A}^{|N|} \mapsto \mathbb{R}$, and the observation function as $O : S \mapsto \Delta(\Omega^1 \times \cdots \times \Omega^{|N|})$.

Each episode in a Dec-POMDP starts from an initial state, $s_0 \in \mathcal{S}$. At timestep $t$, each agent $i \in N$ receives an observation $o_t^i \sim O(s_t)$ and selects an action $a_t^i$ according to its policy $\pi^i(H_t^i)$, which is conditioned on its observation-action history $H_t^i = \left\{ o_{\leq t}^i, a_{<t}^i \right\}$ containing the sequence of observation and actions observed up to

timestep $t$. Agents then jointly execute their selected action in the environment. After execution of the joint action $\boldsymbol{a}_t$, the state of the environment changes according to the transition function $s_{t+1} \sim P(s_t, \boldsymbol{a}_t)$, and each agent is rewarded with $\mathcal{R}(s_t, \boldsymbol{a}_t)$. This reward is common to all agents due to the cooperative nature of AHT problems.

### 3.2 Teammate Policy Generation

A teammate generation process aims to design a set of $K$ teammate policies, $\Pi^{\text{train}} = \{\pi^1, \pi^2, \ldots, \pi^K\}$, that when being used for AHT training maximises the robustness of the learner. Formalising this goal as a quantitative learning objective requires a measure of robustness for a given Dec-POMDP. Once such a robustness measure is formally defined, a learning objective can be formulated by defining how the generated teammate training policies affect the learner's robustness.

We characterise a learner policy as *robust* if it achieves high returns when collaborating with teammates from an unknown evaluation set, $\Pi^{\text{eval}}$. Given a Dec-POMDP where the learner is assigned a fixed index $i \in N$ during the teammate generation, AHT training, and AHT evaluation process, our proposed measure of robustness is defined below:

$$M_{\Pi^{\text{eval}}}(\pi^i) = \mathbb{E}_{\boldsymbol{\pi}^{-i} \sim U(\Pi^{\text{eval}}), a_t^i \sim \pi^i, a_t^{-i} \sim \boldsymbol{\pi}^{-i}, P, O}\left[\sum_{t=0}^{\infty} \gamma^t R(s_t, a_t)\right], \tag{1}$$

with $a_t = \langle a_t^i, a_t^{-i}\rangle$, $\pi^i$, $U(X)$ and $\boldsymbol{\pi}^{-i}$ denote agents' joint action, the policy of the learner, a uniform distribution over a set $X$ and the joint policy of the $|N| - 1$ agents other than the learner respectively. It is important to note that $\Pi^{\text{eval}}$ in Equation 1 may consist of policies not encountered during AHT training, highlighting the need for a robust learner for effective collaboration.

Since the proposed measure of robustness depends on the set of policies in $\Pi^{\text{eval}}$, we outline assumptions regarding the policies that can appear in $\Pi^{\text{eval}}$. As formulated by Stone et al. (2010), we assume that $\Pi^{\text{eval}}$ consists of feasible teammate policies. For $\pi^{-i}$ to be considered feasible, there must be a policy $\pi^i$ that can achieve expected returns above an expert-defined threshold when collaborating with $\pi^{-i}$. Note that this threshold can be decreased if we want to increase the number of feasible policies considered in $\Pi^{\text{eval}}$. Similar to the motivation behind its definition by Stone et al. (2010), the feasibility criteria behind $\pi^{-i} \in \Pi^{\text{eval}}$ reflects how encounters with highly suboptimal teammate policies that no one can collaborate with is improbable in many practical applications of AHT.

As the missing piece to formalise the goal of the teammate generation process, we now define how $\Pi^{\text{train}}$ affects the robustness of a learner produced by AHT methods based on $M_{\Pi^{\text{eval}}}$. Given an AHT method to train a learner, $\Pi^{\text{train}}$ is utilised to learn an optimal AHT policy, $\pi^{*,i}(\Pi^{\text{train}})$ that maximises the expected returns of the learner when collaborating with teammates from $\Pi^{\text{train}}$. Given a Dec-POMDP, the optimal policy given $\Pi^{\text{train}}$ is defined below:

$$\pi^{*,i}(\Pi^{\text{train}}) = \operatorname*{argmax}_{\pi^i} \mathbb{E}_{\pi^{-i} \sim U(\Pi^{\text{train}}), a_t^i \sim \pi^i, a_t^{-i} \sim \boldsymbol{\pi}^{-i}, P, O}\left[\sum_{t=0}^{\infty} \gamma^t R(s_t, a_t)\right]. \tag{2}$$

Later during the AHT evaluation process, $\pi^{*,i}(\Pi^{\text{train}})$ is the policy whose robustness when collaborating with teammates from $\Pi^{\text{eval}}$ will be measured.

Based on the definition of $\pi^{*,i}(\Pi^{\text{train}})$, the goal of a teammate generation process is to find an optimal set of training teammates, $\Pi^{*,\text{train}}$, that maximises the robustness of an AHT agent. Given a Dec-POMDP and an unknown $\Pi^{\text{eval}}$, $\Pi^{*,\text{train}}$ is formally defined as:

$$\Pi^{*,\text{train}} = \operatorname*{argmax}_{\Pi^{\text{train}}} M_{\Pi^{\text{eval}}}\left(\pi^{*,i}(\Pi^{\text{train}})\right). \tag{3}$$

While setting $\Pi^{*,\text{train}} = \Pi^{\text{eval}}$ provides an optimal solution to the above objective, note that the teammate generation problem operates in a setup where $\Pi^{\text{eval}}$ is unknown during training. Therefore, the main challenge in the teammate generation problem arises as a result of optimising for $\Pi^{*,\text{train}}$ without knowing $\Pi^{\text{eval}}$.

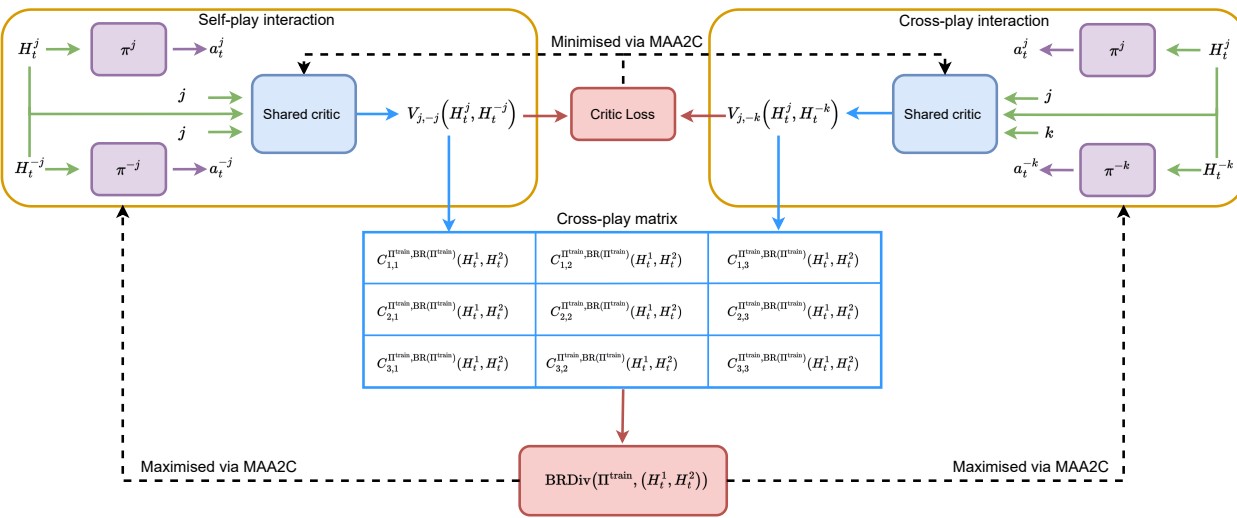

Figure 2: **Teammate Generation By Optimising BRDiv.** This figure visualises our teammate generation method assuming that we are generating $|\Pi^{\text{train}}| = 3$ for AHT environments with two players. Our method utilises MAA2C (Papoudakis et al., 2021b) to generate a set of teammates that maximises the BRDiv diversity metric. The MAA2C algorithm trains a separate *actor network* (purple rectangles) to represent the policies of each generated teammate, $\pi^j \in \Pi^{\text{train}}$, and their associated best-response policies, $\pi^{-j}$. Assuming $\pi^j, \pi^k \in \Pi^{\text{train}}$, a *shared critic network* (green box) is trained to estimate expected returns from the interaction between any possible pairs of $(\pi^j, \pi^{-k})$. The shared critic network's return estimates for all pairs are then compiled into a cross-play matrix (blue bordered box), which serves as a basis to compute the BRDiv diversity metric (red box). Finally, a diverse $\Pi^{\text{train}}$ is produced by optimising the actor networks to maximise the cross-play matrix-based BRDiv metric by minimizing the actor loss outlined in Equation 9.

## 4 Best-Response Diversity Metric

This section provides the details of **B**est-**R**esponse **Div**ersity (BRDiv), the diversity metric that is optimised by our teammate generation method. Section 4.1 starts by outlining a desirable characteristic for $\Pi^{\text{train}}$, a set of teammates policies generated for AHT training. Section 4.2 then formally defines BRDiv as a diversity metric optimised by our teammate generation approach to encourage the creation of a desirable $\Pi^{\text{train}}$.

### 4.1 Desirable Diversity for AHT

A good diversity metric to generate $\Pi^{\text{train}}$ must consider the effect of $\Pi^{\text{train}}$ on a learner's learning process and their robustness when collaborating with policies from $\Pi^{\text{eval}}$. Current AHT methods train a learner to model the encountered teammates and approximate the best-response to each teammate policy in $\Pi^{\text{train}}$ by optimising Equation 2. Although generalisation models used by AHT methods can alleviate this issue, a learner is less likely to achieve high returns when collaborating with $\pi^{\text{eval}} \in \Pi^{\text{eval}}$ whose best-response policy, $\pi^*$ is different to the best-response policy to any $\pi^{\text{train}} \in \Pi^{\text{train}}$. The adverse effects of encountering $\pi^{\text{eval}}$ whose best-response is never learned during training is why setting $\Pi^{\text{train}} = \Pi^{\text{eval}}$ is the ideal solution to maximise learner robustness as defined by Equation 3. However, we often do not know what $\Pi^{\text{eval}}$ is.

Without knowledge regarding $\Pi^{\text{eval}}$, a way to improve a learner's robustness is to increase the number of best-response policies that it learns from interacting with $\pi^{\text{train}} \in \Pi^{\text{train}}$. This idea intuitively aims to reduce the likelihood of a learner being unprepared by not knowing how to best respond to an unknown teammate. Increasing the number of best-response policies learned during training is equivalent to reducing superficial differences between generated teammates, as illustrated in Figure 1.

Our teammate generation method aims to reduce the appearance of redundant policies for AHT training in $\Pi^{\text{train}}$, characterised by their superficial differences. Teammates with superficial differences are indicated by

their common best-response policies. Assuming a sufficiently small number $\epsilon$, a pair of policies $\pi^i, \pi^j \in \Pi^{\text{train}}$ are formally deemed to be superficially different if they share the same best-response policy as defined below:

$$\pi^i \neq \pi^j \ \wedge \left| \mathbb{E}_{a_t^1 \sim \pi^i, a_t^2 \sim \boldsymbol{\pi}^{-i}, P, O} \left[ \sum_{t=0}^{\infty} \gamma^t R(s_t, a_t) \right] - \mathbb{E}_{a_t^1 \sim \pi^i, a_t^2 \sim \boldsymbol{\pi}^{j}, P, O} \left[ \sum_{t=0}^{\infty} \gamma^t R(s_t, a_t) \right] \right| \leq \epsilon,$$

with $a_t = \langle a_t^1, a_t^2 \rangle$ and $\pi^{-i}$ being the best-response policy to $\pi^i$ defined as:

$$\pi^{-i} = \underset{\boldsymbol{\pi}}{\text{argmax}} \ \mathbb{E}_{a_t^1 \sim \pi^i, a_t^2 \sim \boldsymbol{\pi}, P, O} \left[ \sum_{t=0}^{\infty} \gamma^t R(s_t, a_t) \right].$$

Through the interaction with each $\pi^j$ from $\Pi^{\text{train}}$ with minimal superficial differences, the learner will learn as many best-response policies possible to interact with a possible teammate. This equips the learner with a more comprehensive library of behaviours to effectively collaborate with any teammate policy. Consequently, the learner's robustness should improve by reducing the likelihood of it having no adequate strategies to effectively collaborate with an unknown teammate from $\Pi^{\text{eval}}$.

## 4.2 BRDiv Metric

Following our desired characteristic of a diversity metric, this section defines a diversity metric that can be maximised to generate $\Pi^{\text{train}}$ with minimal superficial differences between the generated teammates. The description of BRDiv assumes that only two agents exist in the environment. Assuming a Dec-POMDP where a learner is assigned a fixed index from $N$ during teammate generation, AHT training, and AHT evaluation, extending our proposed diversity metric and optimisation method to environments with more than two agents is straightforward.

BRDiv aims to generate a set of diverse policies for AHT training, $\Pi^{\text{train}} = \{\pi^1, \pi^2, ..., \pi^K\}$, where similar best-response policies cannot be used to effectively collaborate with different generated teammate types from $\Pi^{\text{train}}$. Therefore, defining a metric that quantifies the effectiveness of two agents' policies when collaborating with each other is a crucial first step in formulating our diversity metric. We measure the effectiveness of two policies when collaborating via their expected returns, which is inspired by our notion of robust collaboration introduced in Section 3.2. Assuming that agent $j$ and $k$ are interacting with each other based on policies, $\pi^j(a^j|H_t^j)$ and $\pi^k(a^k|H_t^k)$, that are conditioned on their respective observation-action history $H_t^j$ and $H_t^k$, this return-based effectiveness measure is defined as:

$$V_{j,k}(H_t^j, H_t^k) = \mathbb{E}_{a_T^j \sim \pi^j, a_T^k \sim \pi^k} \left[ \sum_{T=t}^{\infty} \gamma^{T-t} R(s_T, a_T)) \Big| H_t^j, H_t^k \right]. \tag{4}$$

This history-conditioned return-based effectiveness measure provides a foundation for defining an optimised diversity metric to achieve the goal of BRDiv. Denoting the best-response policy to $\pi^k$ by $\pi^{-k,*}$, and the set of best-response policies to each policy in $\Pi^{\text{train}}$ by $\text{BR}(\Pi^{\text{train}})$, we use Equation 4 to evaluate the effectiveness of $\pi^{-k,*} \in \text{BR}(\Pi^{\text{train}})$ when collaborating with $\pi^j \in \Pi^{\text{train}}$. Given a pair of observation-action histories, $H_t^1$ and $H_t^2$, we arrange the measured cooperative effectiveness between all possible $(\pi^j, \pi^{-k,*}) \in \Pi^{\text{train}} \times \text{BR}(\Pi^{\text{train}})$ into a $K \times K$ cross-play matrix, $C^{\Pi^{\text{train}}, \text{BR}(\Pi^{\text{train}})}(H_t^1, H_t^2)$. Elements of this cross-play matrix are defined as:

$$\forall j, k \in \{1, 2, ..., K\}, C_{j,k}^{\Pi^{\text{train}}, \text{BR}(\Pi^{\text{train}})}(H_t^1, H_t^2) = V_{j,(-k,*)}(H_t^1, H_t^2). \tag{5}$$

In this work, the way we compute $C_{j,k}^{\Pi^{\text{train}}, \text{BR}(\Pi^{\text{train}})}(H_t^1, H_t^2)$ and assemble it into the cross-play matrix are illustrated by the blue arrows and table in Figure 2.

The BRDiv metric is based on the intuition that a good $\Pi^{\text{train}}$ to ensure the learner's robustness must possess two characteristics. First, the cross-play matrix of $\Pi^{\text{train}}$ must have high values on its diagonal elements to ensure that each $\pi^j \in \Pi^{\text{train}}$ interacts effectively with its associated best-response policy, $\pi^{-j} \in \text{BR}(\Pi^{\text{train}})$. This characteristic also prevents the emergence of teammate policies producing low returns, which no

reward-optimising agent would have a reason to use in an environment. Second, the off-diagonal elements of $C^{\Pi^{\text{train}},\text{BR}(\Pi^{\text{train}})}$ must also have low values to discourage a best-response policy $\pi^{-j} \in \text{BR}(\Pi^{\text{train}})$ from being effective for collaborating with $\pi^k \in (\Pi^{\text{train}} - \{\pi^j\})$. By optimising the incompatibility of a best-response policy when dealing with other policies in $\Pi^{\text{train}}$, we aim to induce the need for different collaboration strategies to deal with each policy in $\Pi^{\text{train}}$.

Based on these two characteristics, we define our diversity metric as:

$$
\begin{aligned}
\text{BRDiv}(\Pi^{\text{train}}, (H_t^1, H_t^2)) = \ &\text{Tr}\left(C^{\Pi^{\text{train}},\text{BR}(\Pi^{\text{train}})}(H_t^1, H_t^2)\right) \\
&+ \sum_{\substack{i,j\in\{1,\ldots,K\},\\ i\neq j}} \left(C_{i,i}^{\Pi^{\text{train}},\text{BR}(\Pi^{\text{train}})}(H_t^1, H_t^2) - C_{i,j}^{\Pi^{\text{train}},\text{BR}(\Pi^{\text{train}})}(H_t^1, H_t^2)\right) \\
&+ \sum_{\substack{i,j\in\{1,\ldots,K\},\\ i\neq j}} \left(C_{i,i}^{\Pi^{\text{train}},\text{BR}(\Pi^{\text{train}})}(H_t^1, H_t^2) - C_{j,i}^{\Pi^{\text{train}},\text{BR}(\Pi^{\text{train}})}(H_t^1, H_t^2)\right).
\end{aligned}
\tag{6}
$$

The maximisation of the first term in Equation 6 enforces the first characteristic. Meanwhile, maximising the remaining terms produces a cross-play matrix with low off-diagonal values, encouraging the generated policies to fulfil the previously mentioned second desired characteristic.

## 5 Maximising BRDiv with Multi-Agent Reinforcement Learning

We now describe an optimisation technique that maximises BRDiv to generate $\Pi^{\text{train}}$. Although a wide range of multi-agent RL algorithms can be used to maximise BRDiv, we propose an optimisation technique based on the Multi-Agent A2C (MAA2C) algorithm (Papoudakis et al., 2021b) due to the straightforward modifications required to utilise it for maximising BRDiv. We use the centralised critic of MAA2C to estimate the elements of the cross-play matrix defined in Equation 5. Meanwhile, the policies in $\Pi^{\text{train}}$ alongside their associated best-response policies in $\text{BR}(\Pi^{\text{train}})$ are treated as actors that MAA2C trains. A detailed pseudocode of our MARL-based diversity optimisation technique is provided in Algorithm 1 in Appendix C. A visualisation that summarises our proposed teammate generation method is also provided in Figure 2.

**Data Collection.** Before each update to the actors and centralised critic, we separately collects two types of interaction data for training. First, we collect *self-play experiences* where we let a policy, $\pi^k \in \Pi^{\text{train}}$, interact with its associated best-response policy, $\pi^{-k} \in \text{BR}(\Pi^{\text{train}})$. The second type of data is *cross-play experiences* which we collect by letting a policy, $\pi^j \in \Pi^{\text{train}}$, interact with the best-response policy of a different policy, $\pi^{-k} \in \text{BR}(\Pi^{\text{train}} - \{\pi^j\})$. Both self-play and cross-play interaction data are then stored in separate storage denoted by $\mathcal{D}^{\text{SP}}$ and $\mathcal{D}^{\text{XP}}$ respectively. Note that assuming we also record the identity of the agents generating the experience, which is $j$ and $-k$, each experience stored in the storage is then defined as a 7-tuple, $\langle (H_t^1, H_t^2), a_t^j, a_t^{-k}, \{R_t\}, (H_{t+1}^1, H_{t+1}^2), j, -k\rangle$ with $H_t^1$ and $H_t^2$ denoting the observation-action history from using policies $\pi^j$ and $\pi^{-k}$ up to timestep $t$. After the models are updated, $\mathcal{D}^{\text{SP}}$ and $\mathcal{D}^{\text{XP}}$ are emptied and new self-play and cross-play interaction data are collected for the subsequent update.

**Actor and Centralised Critic Architecture.** As we mentioned at the beginning of Section 5, the actors in our optimisation method correspond to the generated teammate policies in $\Pi^{\text{train}}$ and their associated best-response policies. For each $\pi^i \in (\Pi^{\text{train}} \cup \text{BR}(\Pi^{\text{train}}))$, this policy is represented as a neural network parameterised by $\theta^i$. In the remainder of our description of BRDiv, note that we denote the set of actor parameters from $\Pi^{\text{train}} \cup \text{BR}(\Pi^{\text{train}})$ as $\Theta$.

Like the actor networks, the centralised critic used in this optimisation process is also implemented as a neural network. The centralised critic network is specifically responsible for estimating elements of the cross-play matrix, $C^{\Pi^{\text{train}},\text{BR}(\Pi^{\text{train}})}$, based on Equation 5. As shown in Figure 2, the shared critic network input consists of a sequence of observation-action history from both players in the environment. In the remainder of this document, note that we drop $\Pi^{\text{train}}$ as parameters to the cross-play matrix since evaluating each element of this matrix at row $i$ and column $j$ does not involve $\pi^i$ and $\pi^{-j,*}$. Instead, we evaluate $V_{i,-j}^{\phi}(H_t^i, H_t^{-j})$ by also concatenating a one-hot identification of $i$ and $-j$ to the centralised critic's input as indicated by Figure 2.

**Learning Objective.** The centralised critic network is trained to minimise the $n$-step return loss. As in many deep RL methods, we incorporate a target critic network parameterised by $\bar{\phi}$ to compute the target values for the critic network. Using the collected experiences from $\mathcal{D}^{\mathrm{SP}}$ and $\mathcal{D}^{\mathrm{XP}}$, the centralised critic loss function is defined below:

$$L_\phi(\mathcal{D}^{\mathrm{SP}}, \mathcal{D}^{\mathrm{XP}}) = \sum_{\mathcal{D}^{\mathrm{SP}} \cup \mathcal{D}^{\mathrm{XP}}} \frac{1}{2} \left( V_{i,-j}^\phi(H_t^1, H_t^2) - \sum_{k=0}^{n-1} \gamma^k R_{t+k} - \gamma^n V_{i,-j}^{\bar{\phi}}(H_{t+n}^1, H_{t+n}^2) \right)^2. \tag{7}$$

Given a stored experience from $\mathcal{D}^{\mathrm{SP}}$ or $\mathcal{D}^{\mathrm{XP}}$, the actor networks in $\Pi^{\mathrm{train}}$ and $\mathrm{BR}(\Pi^{\mathrm{train}})$ are trained to maximise the BRDiv-based advantage function, $A_{i,-j}^\phi(H_t^1, H_t^2, \{R_{t+k}\}_{k=0}^{n-1}, H_{t+n}^1, H_{t+n}^2)$, defined below:

$$\mathrm{BRDiv}(C_{i,-j}^{\mathrm{pred},\phi}(H_t^1, H_t^2, \{R_{t+k}\}_{k=0}^{n-1}, H_t^1, H_t^2)) - \mathrm{BRDiv}(C^{\mathrm{base},\phi}(s_t)). \tag{8}$$

In the above expression, $C_{i,-j}^{\mathrm{pred},\phi}(H_t^1, H_t^2, \{R_{t+k}\}_{k=0}^{n-1}, H_t^1, H_t^2)$ is a cross-play matrix which has its entry at row $i$ and column $j$ replaced by an n-step return estimate resulting from the interaction between $\pi^i$ and $\pi^{-j}$. Meanwhile, $C^{\mathrm{base},\phi}(s_t)$ is a baseline cross-play matrix whose elements only depend on $H_t^1$ and $H_t^2$.

Similar to its role in the optimisation process of methods based on A3C (Mnih et al., 2016), we use an $n$-step return-based estimate for one of the elements of this cross-play matrix to reduce the bias of gradients associated with the actor loss updates, which is a commonly used method in single-agent actor-critic methods. We then subtract a baseline function from the $n$-step return estimates to reduce the variance of the gradient updates for the actor networks. Finally, note that our $n$-step advantage function estimate highly resembles the advantage function in MAA2C (Papoudakis et al., 2021b), except that we define the advantage function in terms of the best-response diversity of cross-play matrices.

Given stored experiences from $\mathcal{D}^{\mathrm{SP}}$ and $\mathcal{D}^{\mathrm{XP}}$, this results in the use of the following loss function to optimise the actor networks:

$$L_\theta(\mathcal{D}^{\mathrm{SP}}, \mathcal{D}^{\mathrm{XP}}) = \sum_{\mathcal{D}^{\mathrm{SP}} \cup \mathcal{D}^{\mathrm{XP}}} \left( - \log \left( \pi(a_t^i | H_t^1; \theta_i) \pi(a_t^{-j} | H_t^2; \theta_{-j}) \right) \right.$$
$$\left. A_{i,-j}^\phi(H_t^1, H_t^2, \{R_{t+k}\}_{k=0}^{n-1}, H_{t+n}^1, H_{t+n}^2) \right), \tag{9}$$

where,

$$C_{i,-j,p,q}^{\mathrm{pred},\phi}(H_t^1, H_t^2, \{R_{t+k}\}_{k=0}^{n-1}, H_{t+n}^1, H_{t+n}^2) = \begin{cases} V_{p,-q}^\phi(H_t^1, H_t^2), & \text{if } (p,q) \neq (i,j) \\ \sum_{k=0}^{n-1} \gamma^k R_{t+k} + \gamma^n V_{i,-j}^\phi(H_{t+n}^1, H_{t+n}^2), & \text{otherwise} \end{cases}$$
$$C_{m,n}^{\mathrm{base},\phi}(H_t^1, H_t^2) = V_{m,-n}^\phi(H_t^1, H_t^2), \tag{10}$$

are the crosss-play matrices computed to evaluate the advantage function as defined in Equation 8. This objective function that multiplies agents' actions log-likelihood with the advantage function is similar to how actor networks are trained in MAA2C. Minimising Equation 9 updates the actor networks to encourage the emergence of actor networks that assign higher probabilities towards actions leading to trajectories with higher BRDiv values.

## 6 Experiments

We evaluate the effectiveness of BRDiv in improving the robustness of an AHT learner when dealing with previously unseen teammate types. First, we provide details of the environments used in our teammate generation experiments in Section 6.1. This is followed by an overview of our experiments' AHT training and evaluation process in Section 6.2. Section 6.3 then details the baseline approaches we compare BRDiv against. We then present and analyse the results of the teammate generation experiments in Section 6.4. Finally, Section 6.5 analyses the behaviours of teammate types generated by BRDiv.

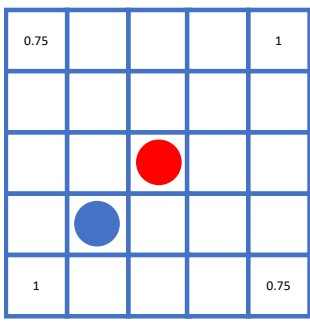

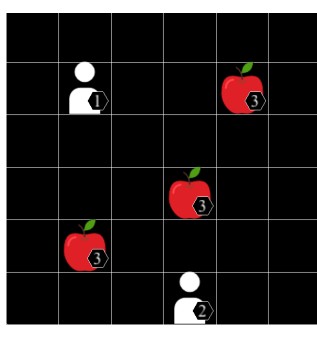

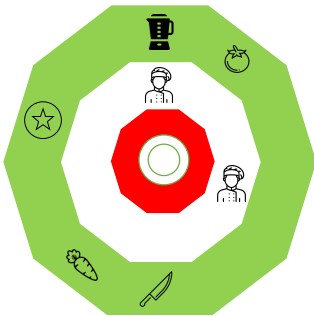

(a) Cooperative Reaching.      (b) Level-Based Foraging.      (c) Simple Cooking.

Figure 3: **Environments for Teammate Generation Experiments.** Figure 3a visualizes an example state of the Cooperative Reaching environment. In this visualisation, the red circle, blue circle, and grids with texts denote the teammate, learner, and reward-providing coordinates. Meanwhile, an example state of level-based foraging environment is visualised in Figure 3b. The white and red icons represent the players and the objects that exist in the environment. The level of each player and object is then visualised in the bottom right corner of their respective icons. Finally, an example environment state for the Simple Cooking environment is provided in Figure 3c. The kitchen layout in this environment is such that the chefs are inside a decagon kitchen with a table in the middle, symbolised by the red decagon with a plate on top of it. The required cooking items and ingredients to finish the recipe are then placed on top of the green counters in this kitchen. To finish the task, all processed ingredients and the plate must be placed on the serving counter which is visualised as a green side of a decagon with a star on top.

## 6.1 Environments

Our experiments evaluate BRDiv and the baseline approaches in three multi-agent environments. All environments used in our experiments have two agents, one of which will be controlled by a teammate policy during an interaction episode. A visualisation of an example state from each environment is shown in Figure 3. Further details of the environments used in our experiments are provided below:

**Cooperative Reaching.** Cooperative reaching is a simple environment situated in a 5×5 grid world. Each agent has five actions corresponding to staying at a particular grid and moving into the four cardinal directions. The goal of all agents is to reach and jointly stay in a grid cell whose location belongs to the set of reward-providing coordinates, $F = \{(0,0), (0,4), (4,0), (4,4)\}$. Within these reward-providing coordinates, $(0,0)$ and $(4,4)$ provide a reward of 1 to both agents once they are in the same grid cell with this coordinate. Meanwhile, the grid in $(0,4)$ and $(4,0)$ only provide a reward of 0.75 once both agents arrive. In this environment, the collaboration strategies correspond to the distinct ways a teammate may select a destination grid within $F$. A robust AHT learner should ideally learn to follow their teammates towards any reward-providing coordinates.

**Level-based Foraging (LBF):** In this environment, agents must retrieve three objects that are randomly scattered in a $6 \times 6$ grid world. Agents can move in either of the four cardinal directions and have a special action that allows them to collect adjacent objects. However, note that agents cannot be positioned in the same grid. At the beginning of each episode, each object and each agent are assigned a level that determines whether an agent may collect an object. To successfully pick up an object, the total level of agents choosing the collection action from grids adjacent to the object must be at least the same as the level of the collected object. We then enforce the need for collaboration between agents by setting the level of each object as the total level of agents in the environment. For every successful collection of an object, agents will then be given a reward of 0.33.

**Simple Cooking:** Simple Cooking is an environment where two chefs must collaborate to create a simple dish with chopped tomatoes and blended carrots. Following Figure 3c, the two chefs can only be positioned

on 10 empty spaces between the cooking counter and the table in the middle of the kitchen. However, they cannot be positioned in the same empty space in the kitchen. Each chef is then equipped with eight actions that enable them to (i) stay still, (ii) move clockwise, (iii) move anti-clockwise, (iv) retrieve an ingredient from a counter, (v) put an ingredient to a counter, (vi) retrieve an ingredient in the middle table, (vii) put an ingredient on the middle table, or (viii) use cooking tools placed on a counter. A chef must be positioned in the space closest to the target counter to collect or put an ingredient from or to a counter. On the other hand, a chef can put or collect items on the table at any time. Using a blender or knife to blend carrots or chop tomatoes requires an agent to be positioned in the space closest to the tool and have the right ingredient placed on the same counter as the tool. In this environment, a reward of 0.25 is provided to both agents right after (i) the tomato is chopped, (ii) the carrot is blended, (iii) both chopped tomato and blended carrot are placed on a plate, and (iv) a plate containing chopped tomatoes and blended carrots has been placed on top of the serving counter.

When deciding the environments used in our experiments, we consider whether the environment has multiple cooperation strategies that do not share the same best response policy for optimal collaboration, which means the learner really needs to adapt to the behaviour of the other agent. All three environments in our work fulfil this criterion as many different teammate strategies can solve the underlying collaboration task. In Cooperative Reaching, multiple strategies correspond to the distinct locations teammates can move towards to get rewarded. The different strategies in LBF correspond to different orderings followed by teammates when collecting the scattered objects. Simple Cooking environment also has different subtask allocation and completion strategies for effective collaboration.

The environments in our experiments are also related to existing environments used for AHT evaluation. Level-based Foraging (Albrecht & Ramamoorthy, 2013; Papoudakis et al., 2021b; Mirsky et al., 2022; Rahman et al., 2021) is a commonly used environment for MARL and AHT evaluation. Simple Cooking is also similar to environments based on the Overcooked game (Wu et al., 2021; Yu et al., 2023). We argue that Simple Cooking is as complex as existing Overcooked environments following the many subtasks that need to be completed by agents, alongside the constricted hallway that restricts the movement of agents.

## 6.2 Experiment Protocol

Our process to evaluate the compared teammate generation methods can be divided into three stages. In the first stage, we run BRDiv and other baseline teammate generation methods to create a set of training teammates $\Pi^{\text{train}}$. The second stage utilises the resulting teammates from the first stage to train an AHT learner. We then evaluate the performance of the robustness of the learner when collaborating with a set of previously unseen teammate types from $\Pi^{\text{eval}}$.

In the first stage, we run each evaluated teammate generation method to produce $K$ teammate types. Each teammate generation method is run for five experiment seeds and learns for $T$ total timesteps. We utilised different $K$ and $T$ for each evaluated environment. In Cooperative Reaching, each teammate generation method is trained for 16 million timesteps to produce four teammate types due to the simplicity of the environment. Meanwhile, each method is trained for 200 million timesteps to produce six and eight different teammate types for LBF and Simple Cooking, respectively. Under each experiment seed, we save $\Pi^{\text{train}}$ produced by each compared algorithm at the end of the teammate generation process.

The second stage utilises $\Pi^{\text{train}}$ generated from the first stage to train a learner policy through AHT training. To enable a fair comparison between results from each teammate generation method, our evaluation protocol uses the same AHT algorithm to train a learner based on each $\Pi^{\text{train}}$. We specifically use the PLASTIC Policy algorithm (Barrett et al., 2017) due to its ease of use for computing a learner policy given $\Pi^{\text{train}}$ produced by our teammate generation methods. In particular, a PLASTIC Policy agent's decision-making process only requires the policy of each teammate and their associated best-response policies, both being a by-product of the teammate generation process contained in the resulting $\Pi^{\text{train}}$ and $\text{BR}(\Pi^{\text{train}})$.

Using the learners produced in the previous stage, the final stage of our experimental protocol evaluates the learner's robustness when dealing with agents from $\Pi^{\text{eval}}$. To evaluate robustness we construct $\Pi^{\text{eval}}$ based on two different scenarios. In the first scenario, $\Pi^{\text{eval}}$ consists of policies generated by the different teammate generation methods evaluated in this work. In the second evaluation scenario, to construct $\Pi^{\text{eval}}$

Table 1: **Experiment Baselines.** This table outlines the differences between our method and the baselines compared in our experiments. The comparison between these different teammate generation methods is based on their optimised loss functions, self-play and cross-play data for training, and the use of a policy classifier to produce intrinsic rewards for MAA2C training.

| Method | Loss Function | Self-Play Data | Cross-Play Data | Policy ID Classifier |
|---|---|---|---|---|
| BRDiv (Our method) | Equations 7 & 9 | Yes | Yes | No |
| Independent | Equations 7 & 9 | Yes | No | No |
| TrajeDi | Equations 7 & 9 with auxiliary loss | Yes | No | No |
| Any-Play | Equations 7 & 9 with intrinsic rewards | Yes | No | Yes |

we define policies based on the predefined heuristics (defined in Appendix B). Note that when evaluating against teammates generated by the same teammate generation method in the first scenario, we also measure the learner's robustness against teammate policies generated from different teammate generation experiment seeds. Such evaluation remains challenging since teammates generated by the same method under different seeds may have different behaviours that cause difficulties in effective collaboration.

As a measure of robustness, our evaluation process proceeds by evaluating the returns of an AHT learner when dealing with teammate policies in $\Pi^{\text{eval}}$. For each environment and evaluation scenario, we then compute an aggregated statistic of the returns achieved by each method when dealing with policies from $\Pi^{\text{eval}}$. Using the stratified bootstrap confidence interval method (Agarwal et al., 2021), we compute a 95% confidence interval over the interquartile mean returns of AHT learners resulting from training with $\Pi^{\text{train}}$ from each teammate generation method. This confidence interval allows us to argue over the significance of the difference in robustness between teammate generation methods. We then also report a breakdown of the mean returns achieved by the learner across the different policy types in $\Pi^{\text{eval}}$. The resulting returns of a learner trained through generated teammate types produced by BRDiv and baseline approaches are reported and analysed in Section 6.4.

### 6.3 Baselines

We compared our proposed method with two types of baselines. The first type of baseline comprises previous methods for automatically generating teammates in AHT or related problems, such as zero-shot coordination. Meanwhile, the second type of baseline consists of an ablation of our method, which removes parts of it responsible for encouraging ineffective collaboration between a generated teammate policy and the best-response policy associated with another generated teammate type. Since our experiments operate on fully observable environments, our loss functions are optimised such that Equations 7 and 9, auxiliary loss functions, and intrinsic rewards optimized by the compared methods are defined over the state of the environment. Further details of these methods and their implementation are summarised in Table 1 and the remainder of this section. Appendix C then provides the value of each methods' hyperparameters used in our experiments.

**Prior teammate generation methods.** Among methods under this category, we choose TrajeDi (Lupu et al., 2021) and Any-Play (Lucas & Allen, 2022) as representative baselines. We choose TrajeDi following its usage of the *action discounting term*, which provides additional flexibility when defining the optimised information-theoretic diversity metric. Prior teammate generation methods other than TrajeDi define their optimised diversity metric in terms of an agent's overall trajectory or its selected action at each timestep, which both have their drawbacks. TrajeDi's action discounting term enables users to tune the resemblance of its optimised diversity metric to an action diversity and trajectory diversity-based approach. In the plots that we report in this work, we denote these TrajeDi-based baselines as **TrajeDi0**, **TrajeDi025**, **TrajeDi05**, **TrajeDi075**, and **TrajeDi1**, which use the action discounting term of 0, 0.25, 0.5, 0.75, and 1 respectively.

We also add Any-Play as a baseline following the results from Lucas & Allen (2022) that demonstrated its improved performance over TrajeDi in a few environments. This baseline will be denoted in our analysis as **AnyPlay**. Unlike BRDiv and TrajeDi, Any-Play's teammate generation process adds an intrinsic reward that the actor networks also attempt to maximise aside from the original rewards from the environment.

This intrinsic reward specifically evaluates the log-likelihood assigned by a classifier that distinguishes the different policies in $\Pi^{\text{train}}$. Intuitively, Any-Play optimizes different actors to produce trajectories that are distinguishable from each other. Thus, comparing BRDiv's performance against Any-Play also delivers insights regarding the gains from using a different optimisation technique to induce diversity.

We implement TrajeDi and Any-Play based on our MAA2C-based teammate generation method by first removing the evaluation of loss functions based on cross-play data. Compared to our proposed approach, the absence of any training based on cross-play data prevents TrajeDi and Any-Play from minimizing the non-diagonal elements of the cross-play matrices defined in Equation 10. Maximising Equations 7 and 9 solely based on self-play data has the effect of encouraging each generated policy in $\Pi^{\text{train}}$ to achieve the highest possible returns when interacting with its best-response policy.

TrajeDi and Any-Play also add additional auxiliary losses or intrinsic rewards to encourage diversity in $\Pi^{\text{train}}$. Using $\mathcal{D}^{\text{SP}}$ for its evaluation, TrajeDi adds an auxiliary loss that minimises the negative Jensen-Shannon Divergence between the trajectory distributions of the actor networks being trained. For Any-Play, we add a loss function that trains a classifier that identifies the population a teammate belongs to based on an observed state and its action. The output of this classifier is then used at each timestep to compute an intrinsic reward that is added on top of the environment rewards during training.

**Ablations of BRDiv.** We also compare our method against an ablation which independently trains $K$ teammate policies with MAA2C (Papoudakis et al., 2021b) without maximising any policy diversity metrics. Our experiments denote this baseline as **Independent**. Comparing our method's performance against this ablation helps us identify the impact of optimising our proposed diversity metric on the resulting learner's robustness when dealing with previously unseen teammates. We train this ablation to maximise Equations 7 and 9 solely based on self-play data, which similar to TrajeDi and Any-Play encourage the generated policy to maximise their returns against their respective best-response policies. However, we do not add any intrinsic rewards or optimise auxiliary losses to encourage diversity among the generated teammate policies.

## 6.4 AHT Evaluation

This section provides the results of using the generated teammate policies for training an AHT learner based on the experimental protocol outlined in Section 6.2. The aggregated performance of learners trained with $\Pi^{\text{train}}$ produced by each teammate generation method is outlined in Figure 4. Figure 5 show the performance of a PLASTIC Policy-based learner when interacting with teammates that follow one of the previously unseen heuristics defined in Appendix B. The performance of the same learner when dealing with previously unseen teammates generated by different teammate generation methods is then provided in Figure 6.

As shown in Figures 4a, 4b, 5a and 5b, BRDiv provides a more reliable way to generate robust learners than the baseline methods when collaborating with unseen heuristics that are near optimal. When comparing the resulting returns between BRDiv-based learners with the Independent baseline, we see that BRDiv achieves higher returns in all but three heuristics throughout the entire types of teammates used in the evaluation process. Except for interactions against teammates using heuristics H08 and H10, a learner trained with BRDiv-based teammates consistently achieves the highest average returns compared to the other evaluated teammate generation methods in Cooperative Reaching. Meanwhile, BRDiv also consistently yields more robust learners than compared baselines in LBF except for interactions against teammates using heuristic H02. In experiments against teammates using policies generated by other teammate generation methods whose results are provided by 4d- 4f and 6a- 6c, learners trained using BRDiv-based teammates consistently achieve the highest average returns compared to other baseline methods in all environments except for Simple Cooking. Finally, note that in cases where specific baseline methods outperform a BRDiv-based learner in terms of the resulting average returns, the difference in performance between BRDiv and these baselines is insignificant except for some evaluation scenarios based on Simple Cooking.

Another substantial evidence of BRDiv's reliability in training robust learners can be found by comparing the confidence interval of returns between compared methods. In Figures 4, 5 and 6, we observe BRDiv's tendency to produce more compact confidence intervals in its returns indicates lower variance in a BRDiv-based learner's returns across different training seeds. The baseline methods' larger variance in returns results from their generated $\Pi^{\text{train}}$ having high variance in best-response diversity across different experiment seeds.

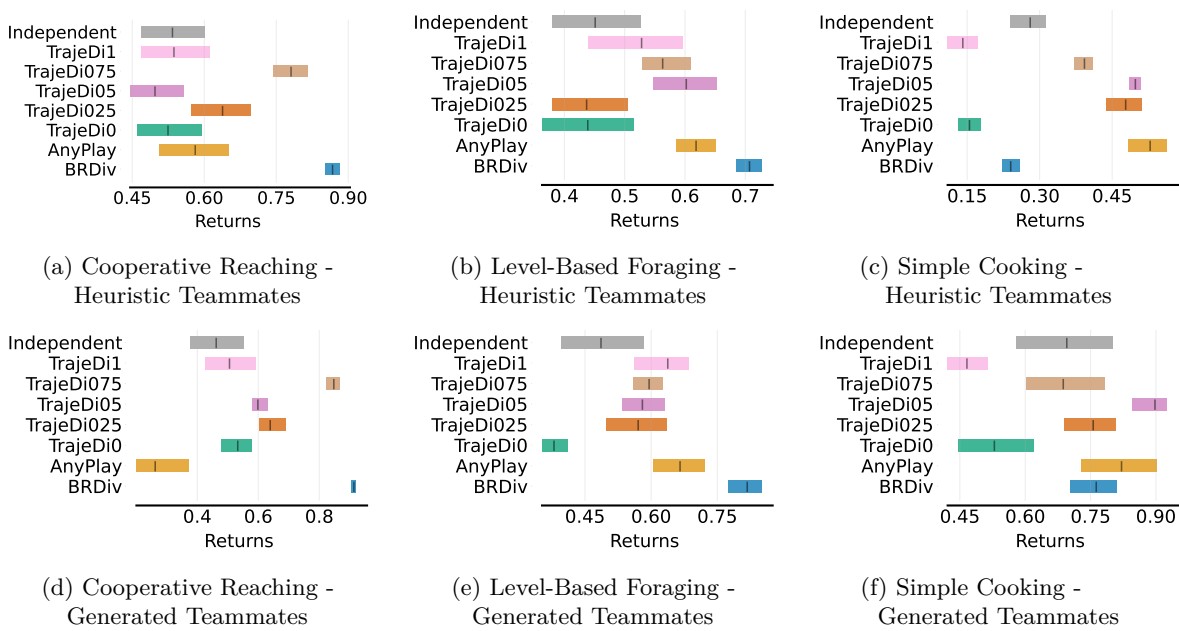

(a) Cooperative Reaching -
Heuristic Teammates

(b) Level-Based Foraging -
Heuristic Teammates

(c) Simple Cooking -
Heuristic Teammates

(d) Cooperative Reaching -
Generated Teammates

(e) Level-Based Foraging -
Generated Teammates

(f) Simple Cooking -
Generated Teammates

Figure 4: **Aggregated Learner Performance During Collaboration With Policies From $\Pi^{\text{eval}}$.**
This figure visualises an aggregate statistic of the episodic returns achieved by a learner trained with $\Pi^{\text{train}}$
when collaborating with policies from $\Pi^{\text{eval}}$. We show results for each baseline and evaluation scenario
detailed in Sections 6.1 and 6.2. The learner interacts with each teammate type that constitutes $\Pi^{\text{eval}}$ for five
episodes. Using the stratified bootstrap confidence interval method (Agarwal et al., 2021), we report the 95%
confidence interval of the interquartile mean of episodic returns the learner achieves. Results show that our
method significantly outperforms all baselines across the two evaluation scenarios in Cooperative Reaching
and Level-Based Foraging. Meanwhile, our method obtained lower returns than the baselines in the Simple
Cooking environment against heuristics.

Across some seeds, the baseline methods still discover $\Pi^{\text{train}}$ with a high BRDiv value even without optimising
BRDiv. These baseline methods can discover $\Pi^{\text{train}}$ with high BRDiv values since policies in $\Pi^{\text{train}}$ with high
BRDiv also exhibit high diversity in the trajectories they generate. However, since high trajectory diversity
does not imply high BRDiv, a few seeds of the baseline methods also discover $\Pi^{\text{train}}$ with lower BRDiv that
yields learners with lower returns due to superficial differences between generated policies. When comparing
the BRDiv value of $\Pi^{\text{train}}$ by different teammate generation methods and the learner's episodic returns
in Cooperative Reaching and LBF against heuristics in $\Pi^{\text{eval}}$, we found these measures to have a strong
Pearson correlation coefficient of 0.7664 and 0.7961 respectively. While TrajeDi has an action discounting
hyperparameter that can be tuned to minimise the emergence of $\Pi^{\text{train}}$ with superficial differences (Lupu
et al., 2021), our results indicate that tuning this hyperparameter is less effective in preventing the emergence
of superficial differences between generated teammates compared to directly optimising BRDiv.

Important insights are also obtained from evaluating learners when collaborating with more suboptimal
teammates. Against teammate-following heuristics H08 and H10 in Cooperative Reaching, BRDiv ceased to
become the best-performing teammate generation method to improve the robustness of the learner. The
same trend is seen in Figure 5c where the learner must collaborate with H01-H12 whose expertise only spans
parts of tasks in the environment, such as processing the ingredients, assembling them into a dish, and
delivering it to a serving counter. This echoes with the results of BRDiv when dealing with other Simple
Cooking teammate policies generated by other baseline teammate generation methods, which we discuss
in Section 6.5 to have generated suboptimal policies. All these results point towards the inadequacy of
BRDiv-based generated policies to improve learner robustness when dealing with suboptimal teammates.

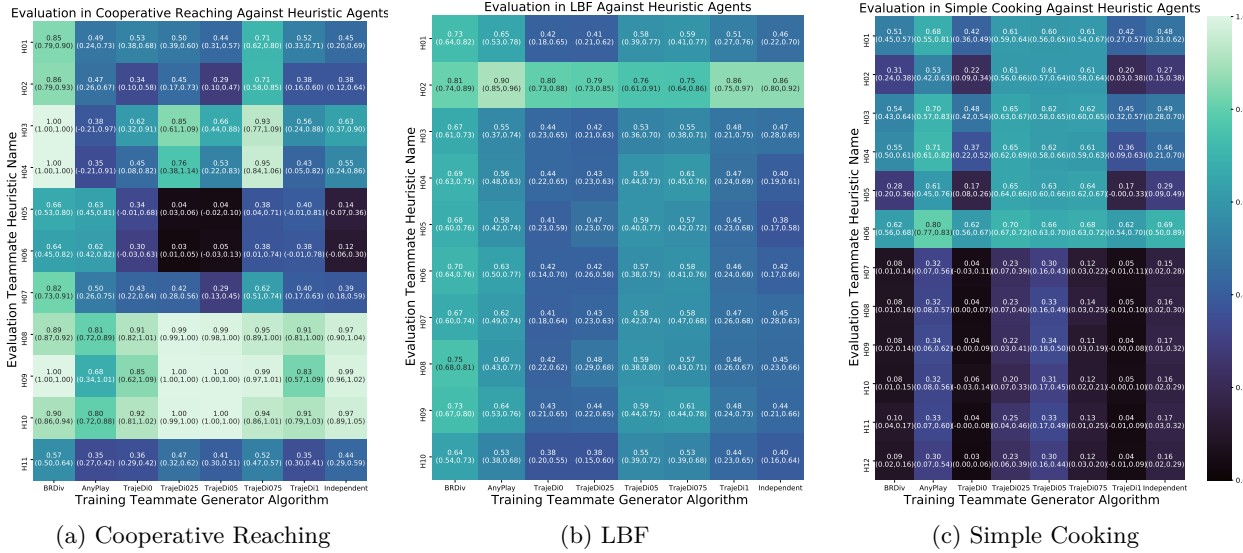

Figure 5: **AHT Evaluation Results Against Heuristic-based Teammates.** We provide the average returns resulting from the interaction between $\Pi^{\text{eval}}$ consisting of heuristic-based teammates and the learner, which is trained using PLASTIC Policy (Barrett et al., 2017) and $\Pi^{\text{train}}$ produced by the evaluated teammate generation methods. Labels on the x-axis of the heatmap visualisation indicate the teammate generation method used to produce $\Pi^{\text{train}}$. Labels on the y-axis highlight the heuristics followed by agent policies from $\Pi^{\text{eval}}$. Within each entry of the heatmap, the first number provides the average returns from the collaboration between learners trained with $\Pi^{\text{train}}$ generated by the method indicated in the x-axis and teammates following heuristics labelled in the y-axis. The numbers in the parentheses provide a 95% confidence interval of the returns based on teammate generation experiments conducted across five seeds. Figure 5a show the results in the Cooperative Reaching environment where training a learner with BRDiv-based teammates produces more robust agents that can deliver higher returns than the baselines, except for interactions against H08 and H10. Meanwhile, the LBF environment results also mirror the findings from the Cooperative Reaching environment, where a BRDiv-based learner yields higher average returns than all baselines except for interactions against H02. Finally, Figure 5c show the Simple Cooking environment results where BRDiv did not achieve the best performance compared to other methods due to its inadequacy when dealing with suboptimal teammate heuristics.

## 6.5   Behaviour Evaluation

In this section, we provide additional empirical evidence regarding the effectiveness of BRDiv in generating $\Pi^{\text{train}}$ for AHT training. First, we show an example of $\Pi^{\text{train}}$ exhibiting superficial differences based on the results of running one of our baseline teammate generation methods in the Cooperative Reaching environment. We then show how BRDiv successfully avoids generating $\Pi^{\text{train}}$ exhibiting superficial differences, which then leads to improved learner robustness when $\Pi^{\text{train}}$ is used for AHT training.

An example of $\Pi^{\text{train}}$ with superficial differences discovered by one of our baseline methods can be seen in Figure 7a. In this visualisation, multiple policies in $\Pi^{\text{train}}$ move towards the same reward-providing coordinates. Effective collaboration with these policies can be achieved through the same best-response policy of moving towards a reward-providing grid the teammate moves towards. This commonality in best-response policies is reflected in Figure 7b, which shows the cross-play matrix resulting from the interaction between the policies in $\Pi^{\text{train}}$ and BR($\Pi^{\text{train}}$).

Training a Cooperative Reaching learner based on $\Pi^{\text{train}}$ in Figure 7a will not provide a robust learner. This is because certain teammate behaviours are not present in the training set, such as teammates that move towards the upper-left or bottom-right reward-providing corners. A better $\Pi^{\text{train}}$ for training learners in Cooperative Reaching is visualised by the BRDiv-based teammate policies in Figure 8a. In this case, $\Pi^{\text{train}}$

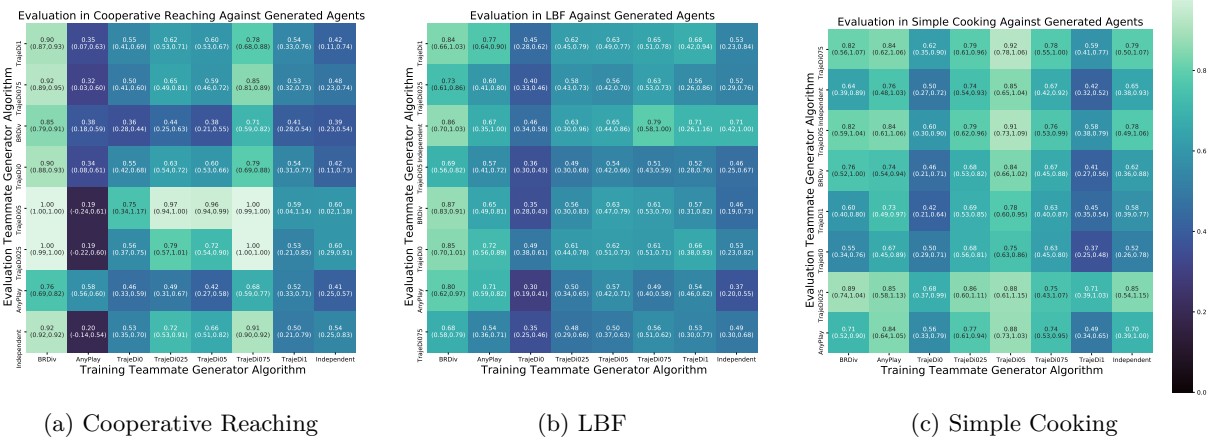

(a) Cooperative Reaching          (b) LBF          (c) Simple Cooking

Figure 6: **AHT Evaluation Results Against Previously Unseen Generated Teammates.** Given $\Pi^{\mathrm{train}}$ generated by a teammate generation method, we also report the average returns achieved by the learner when dealing with $\Pi^{\mathrm{eval}}$ consisting of teammates generated by the different evaluated teammate generation methods. In the figures above, the labels on the heatmap's x-axis, y-axis, and numbers have similar semantics with their respective counterparts in Figure 5. Note that when dealing with $\Pi^{\mathrm{eval}}$ generated by the same algorithm producing $\Pi^{\mathrm{train}}$, it is possible that effective collaboration cannot be achieved since $\Pi^{\mathrm{eval}}$ also consists of policies generated through experiments using different seeds from which is being used to produce $\Pi^{\mathrm{train}}$. In Cooperative Reaching and LBF, BRDiv-based learners produce higher average returns when dealing with previously unseen teammates. For certain teammate generation methods, the difference between a BRDiv-based learner's and its mean returns is even statistically significant. For Simple Cooking, our method occasionally struggles to deal with suboptimal teammate policies generated by some teammate generation methods. Further discussions regarding the suboptimality of policies produced by certain methods are provided in Section 6.5.

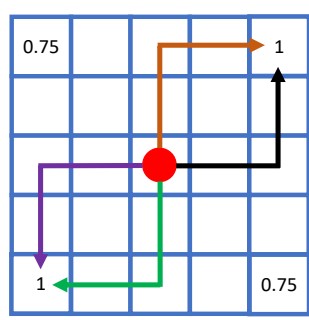

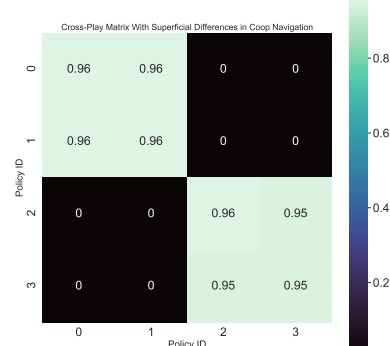

(a) Example of superficial differences between generated teammates.

(b) Cross-play matrix between policies generated following Figure 7a.

Figure 7: **Example of Superficial Policy Differences in Cooperative Reaching.** From one of the $\Pi^{\mathrm{train}}$ resulting from a baseline teammate generation method in our experiments, we see an example of teammates with superficial differences in Cooperative Reaching. Figure 7a show that superficial difference is characterised by different teammate policies that move a teammate towards the same reward-providing corner. Since an effective collaboration with teammates having superficial differences can be achieved using the same best-response policy, the cross-play matrix from Figure 7b demonstrates the compatibility of some best-response policies with multiple policies from $\Pi^{\mathrm{train}}$.

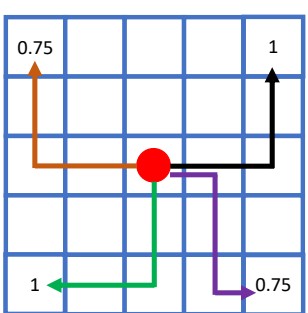

(a) $\Pi^{\mathrm{train}}$ generated by BRDiv in Cooperative Reaching.

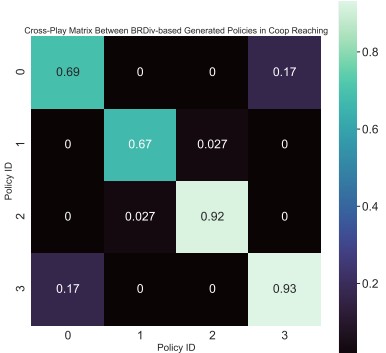

(b) Cross-play matrix between the policies generated by BRDiv in Figure 8a.

Figure 8: **Ideal $\Pi^{\mathbf{train}}$ for Cooperative Reaching.** An example $\Pi^{\mathrm{train}}$ generated by maximising BRDiv is provided in Figure 8a. Compared to the teammate policies displayed in Figure 7a, a robust learner is more likely to be produced from training with this $\Pi^{\mathrm{train}}$ since it contains different policies that move teammates towards all the reward-providing coordinates in Cooperative Reaching. Teammate policies that move towards the upper left and bottom right corners are specifically better handled if the learner trains against policies visualised in Figure 8a. Following Figure 8b, this $\Pi^{\mathrm{train}}$ is characterised by the distinct best-response policies required for effective collaboration against each generated policy.

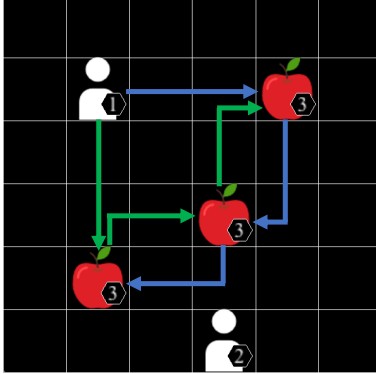

(a) Trajectories produced by two randomly sampled policies from $\Pi^{\mathrm{train}}$.

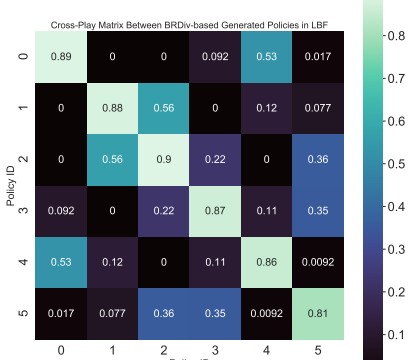

(b) Cross-play matrix from $\Pi^{\mathrm{train}}$ produced by BRDiv.

Figure 9: **$\Pi^{\mathbf{train}}$ Generated by Optimising BRDiv in LBF.** Assuming that the level one agent is the teammate, example trajectories from two randomly sampled policies generated by optimising BRDiv for LBF are displayed in Figure 9a. In this visualisation, each sequence of the same coloured arrows starting from the teammate's position corresponds to a trajectory of a single policy from $\Pi^{\mathrm{train}}$. Different policies in $\Pi^{\mathrm{train}}$ specifically correspond to the distinct orderings that a teammate may follow to collect objects in the environment. Since an effective collaboration with teammates that follow a specific object collection ordering requires best-response policies that follow the same object collection ordering, the best-response to every policy in $\Pi^{\mathrm{train}}$ is distinct and incompatible for collaboration with other policies. This results in the cross-play matrix displayed in Figure 9b.

produces a more robust learner by equipping it with a more comprehensive set of strategies against teammates moving towards any reward-providing grid. We can consistently find this desirable $\Pi^{\mathrm{train}}$ since each policy in $\Pi^{\mathrm{train}}$ requires different best-response policies, which makes it highly likely to be discovered by optimising BRDiv.

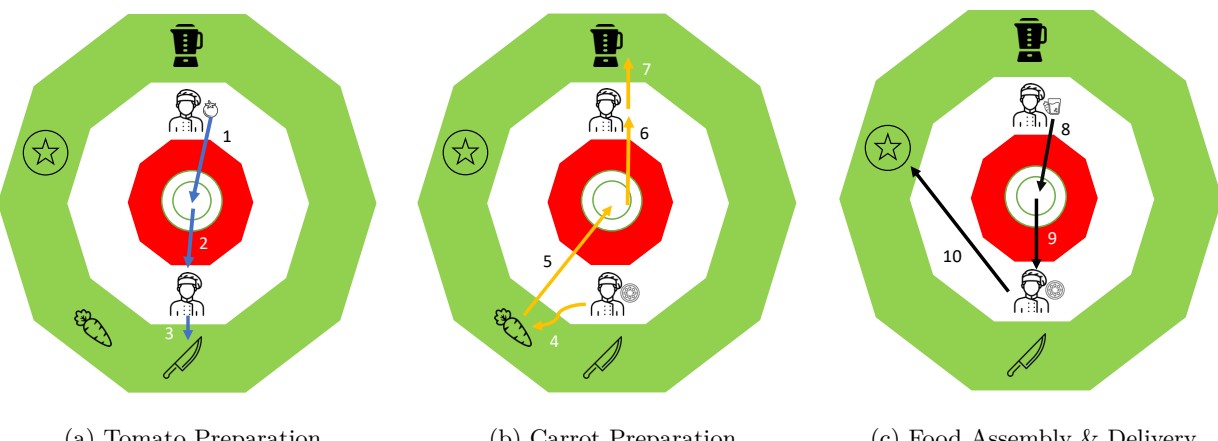

(a) Tomato Preparation.   (b) Carrot Preparation.   (c) Food Assembly & Delivery.

Figure 10: **Teammate Policy Generated By Optimising BRDiv and Its Best-Response Policy.** From left to right, we show an example of a teammate policy alongside its best-response policy generated by optimising BRDiv. The generated policy and its best-response policy learn to quickly divide the ingredient preparation, dish assembly, and delivery tasks between themselves. In Figure 10a, the generated policy collects the tomato and puts it on the table so another teammate closer to the knife and positioned on the opposite side of the kitchen can retrieve and chop it. After chopping the tomato, Figure 10b then shows that the best-response policy learns to move towards the carrot and puts it on the table so that the agent controlled by the generated policy can collect and blend it. After the generated policy puts the blended carrot on the table, the best-response policy collects the plate and carrot to combine it with its chopped tomatoes. The best-response policy eventually delivers this combined food to the serving counter as seen in Figure 10c.

Optimising BRDiv also enables the discovery of a $\Pi^{\text{train}}$ that encourages the emergence of robust learners in the LBF environment. As seen in Figure 9a, each policy in $\Pi^{\text{train}}$ generated by optimising BRDiv for LBF corresponds to the distinct orderings that an optimal agent may take to collect objects in the environment. Since any optimal or near-optimal teammate should follow one of the six possible orderings when collecting objects, the discovery of $\Pi^{\text{train}}$ containing policies that follow each ordering prevents the learner from not having an adequate strategy to deal with optimal or near-optimal teammates. As in the case with Cooperative Reaching, note that the discovery of good quality teammate policies for LBF is made possible by each policy in $\Pi^{\text{train}}$ requiring different best-response policies, which makes it likely to be discovered by optimising BRDiv.

Besides highlighting why optimising BRDiv facilitates improved learner robustness in Cooperative Reaching and LBF, analysing the behaviour of policies generated by our method and the baselines also provides insights into why our method does not yield the most robust learner in Simple Cooking. As displayed by Figure 10, teammate policies generated by optimising BRDiv are highly optimal. The generated policies alongside its best-response policy quickly learn to divide and execute the available subtasks among themselves. Between different generated policies in $\Pi^{\text{train}}$, a difference emerges due to different task assignments between agents and different orderings to complete the subtasks. In general, BRDiv-based generated policies and their best-response policies tend to finish an episode of Simple Cooking in 17-20 timesteps. Learning from such highly optimal policies makes a learner unprepared when facing highly suboptimal policies during evaluation.

This result from BRDiv highly contrasts with the results from optimising alternative diversity metrics tested in this work. A detailed breakdown of the number of timesteps required by each compared teammate generation method to solve Simple Cooking in self-play is provided in Table 2. The better-performing baselines for generating robust learners for Simple Cooking against heuristic-based teammates particularly produce suboptimal teammate policies that solve the environment in 30-195 timesteps. Throughout interaction, Any-Play and TrajeDi-based policies often exhibit suboptimal behaviour such as (i) going back and forth between putting an item on the counter and retrieving it again or (ii) stopping working on subtasks and doing nothing. The availability of such suboptimal policies in $\Pi^{\text{train}}$ makes the learner more prepared to

Table 2: **Required Timesteps to Solve Simple Cooking in Self-Play.** The number of timesteps required by each method to solve Simple Cooking is provided in their respective entries in the second row.

| BRDiv | Any-Play | TrajeDi0 | TrajeDi025 | TrajeDi05 | TrajeDi075 | TrajeDi1 | Independent |
|-------|----------|----------|------------|-----------|------------|----------|-------------|
| 17-20 | 30-35 | 18-25 | 65-158 | 85-233 | 53-161 | 20-28 | 18-29 |

complete a task on its own in case teammates are performing poorly in the task. While this inability to deal with highly suboptimal teammates presents potential research directions to further improve BRDiv, note that such suboptimal teammates are rarely encountered in many realistic applications of AHT. As originally formulated by (Stone et al., 2010), encountered teammates are normally assumed to be capable of achieving a specific return threshold at the given task.

## 7    Conclusion & Future Work

In this work, we discussed the importance of generating a collection of training teammate policies, $\Pi^{\text{train}}$, that require different best-response policies to improve the robustness of an AHT agent. To achieve this, we proposed a teammate generation method that optimises BRDiv, a diversity metric designed to prevent the emergence of superficial differences between policies in $\Pi^{\text{train}}$. Based on a comparison against TrajeDi (Lupu et al., 2021), Any-Play (Lucas & Allen, 2022), and a baseline that independently trains different teammate policies via MARL, our experiments show that optimising BRDiv achieves higher average returns when dealing with near-optimal previously unseen teammate policies. At the same time, we also see a smaller variance in the returns achieved by learners trained with $\Pi^{\text{train}}$ produced by optimising BRDiv.

The conducted analysis of the generated teammates' behaviour showed that optimising BRDiv avoids generating teammates with superficial differences. At the same time, $\Pi^{\text{train}}$ generated by optimising BRDiv covers a comprehensive set of reward-maximising teammate behaviours. Training against this set of teammates eventually produced teammates that can perform a wider range of strategies to collaborate against previously unseen teammate policies.

Although our results in the teammate generation experiments show that optimising BRDiv can generate teammate policies that require different strategies for effective collaboration, we note that this is not the only type of diversity displayed by decision-making agents in real-world problems. In many applications of AHT, a learner also has to deal with teammates that vary in their ability to maximise the teams' returns. For example, even with different teammates that prefer a specific role such as being a striker, we see a wide range of skill levels between potential teammates in a pick-up soccer game. A teammate's ability may range from having the skills of an amateur player to possessing elite skills displayed by top-division professional players. Currently, this diversity cannot be discovered solely based on optimising BRDiv. The first term on the right-hand side of Equation 6 encourages the creation of teammates with near-optimal policies when we optimise BRDiv. By only training a learner against teammates generated by optimising BRDiv, this limitation potentially results in a learner yielding suboptimal returns when dealing with teammates with a low skill level. The results of our experiments in the Simple Cooking environment also confirmed the need for further developments in this direction.

The proposed method to optimise BRDiv also faces challenges when dealing with problems other than two-player games. In many real-world problems such as those addressed in open ad hoc teamwork (Rahman et al., 2021), generating a team of multiple agents with different policies is desirable. While our proposed optimisation method can be modified to generate a team of teammates, many such teams must be generated at once to improve the robustness of the learner. After all, the number of generated training teams must match the exponential increase in the space of possible team configurations. Since training agents via MARL may require millions of experiences even in simple domains, the computational resources required by our proposed method to generate a large collection of teams can quickly grow impractical as the size of a generated team increases.

**Acknowledgments**

This research received financial support from various sources. A.R. received funding from the Edinburgh Enlightenment scholarship. E.F. was supported by the United Kingdom Research and Innovation (grant EP/S023208/1), EPSRC Centre for Doctoral Training in Robotics and Autonomous Systems (RAS). I.C. and S.A. were recipients of funding from the US Office of Naval Research (ONR) via grant N00014-20-1-2390, and the Google Cloud Research Credits program award.

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

## A  BRDiv Pseudocode

We complete the description of our method by providing a pseudocode for the teammate generation process undergone in BRDiv, shown in in Algorithm 1. An essential part of Algorithm 1 is a call to the **COMPUTE_LOSS** function that evaluates the loss functions minimised by BRDiv. How BRDiv utilises the gathered self-play and cross-play experience to compute the minimised loss functions is then described in Algorithm 2.

## B  Heuristic-based Teammates

As we mentioned in Section 6.2, we use $\Pi^{\text{eval}}$ consisting of heuristic-based policies to evaluate the methods used in our experiments. The details of heuristics followed by each policy for the Cooperative Reaching

---

**Algorithm 1** BRDiv-based Teammate Generation Process

---

**Require:**

    Number of training episodes, $n_{\text{eps}}$.

    Episode length, $T$.

    Update period, $t_{\text{update}}$.

    Number of generated teammate types, $K$.

    Initial population actor network parameters, $\Theta = \{\theta_1, \theta_2, ..., \theta_K, \theta_1^{\text{BR}}, \theta_2^{\text{BR}}, ..., \theta_K^{\text{BR}}\}$.

    Initial centralised critic parameters, $\phi$.

    Target centralised critic parameters, $\bar{\phi}$.

    Learning Rate, $\alpha$.

    Target network update coefficient, $\bar{\alpha}$.

    Environment for SP and XP interaction, $\text{env}^{\text{SP}}$ & $\text{env}^{\text{XP}}$.

  1: **for** $i = 1$ to $n_{eps}$ **do**

  2:    $t \leftarrow 0$

  3:    $\mathcal{D}^{\text{SP}}, \mathcal{D}^{\text{XP}} \leftarrow \{\}, \{\}$

  4:    $\text{ID}^{\text{SP}} \sim \text{Uniform}(\{1, \ldots, K\})$                   ▷ Sample Population ID for SP

  5:    $\text{ID}_1^{\text{XP}}, \text{ID}_2^{\text{XP}} \sim \text{Uniform}(\{(i,j)|i,j \in 1, \ldots, K, i \neq j\})$     ▷ Sample Population ID for XP

  6:    Observe $\boldsymbol{H}_0^{SP} = (o_0^{1,\text{SP}}, o_0^{2,\text{SP}})$ and $\boldsymbol{H}_0^{\text{XP}} = (o_0^{1,\text{XP}}, o_0^{2,\text{XP}})$ from $\text{env}^{\text{SP}}$ and $\text{env}^{\text{XP}}$ respectively.

  7:    $H_0^{1,SP}, H_0^{2,SP}, H_0^{1,XP}, H_0^{2,XP} \leftarrow \{o_0^{1,\text{SP}}\}, \{o_0^{2,\text{SP}}\}, \{o_0^{1,\text{XP}}\}, \{o_0^{2,\text{XP}}\}$

  8:    **while** $t < T$ **do**

  9:        // Self-Play Data Collection

10:        $a_t^{1,\text{SP}} \sim \pi\left(a_t^{1,\text{SP}}|H_t^{1,\text{SP}}; \theta_{\text{ID}^{\text{SP}}}\right)$ and $a_t^{2,\text{SP}} \sim \pi^{\text{BR}}\left(a_t^{2,\text{SP}}|H_t^{2,\text{SP}}; \theta_{\text{ID}^{\text{SP}}}^{\text{BR}}\right)$

11:        $r_{t+1}^{\text{SP}}, \boldsymbol{H}_{t+1}^{\text{SP}} \leftarrow \text{env}^{\text{SP}}\left(\boldsymbol{H}_t^{\text{SP}}, \boldsymbol{a}_t^{\text{SP}}\right)$

12:        $\mathcal{D}^{\text{SP}} \leftarrow \mathcal{D}^{\text{SP}}||\langle\boldsymbol{H}_t^{\text{SP}}, \boldsymbol{a}_t^{\text{SP}}, r_{t+1}^{\text{SP}}, \boldsymbol{H}_{t+1}^{\text{SP}}, \text{ID}^{\text{SP}}\rangle$

13:        // Cross-Play Data Collection

14:        $a_t^{1,\text{XP}} \sim \pi\left(a_t^{1,\text{XP}}|H_t^{1,\text{XP}}; \theta_{\text{ID}_1^{\text{XP}}}\right)$ and $a_t^{2,\text{XP}} \sim \pi^{\text{BR}}\left(a_t^{2,\text{XP}}|H_t^{2,\text{XP}}; \theta_{\text{ID}_2^{\text{XP}}}^{\text{BR}}\right)$

15:        $r_{t+1}^{\text{XP}}, \boldsymbol{H}_t^{\text{XP}} \leftarrow \text{env}^{\text{XP}}\left(\boldsymbol{H}_t^{\text{XP}}, \boldsymbol{a}_t^{\text{XP}}\right)$

16:        $\mathcal{D}^{\text{XP}} \leftarrow \mathcal{D}^{\text{XP}}||\langle\boldsymbol{H}_t^{\text{XP}}, \boldsymbol{a}_t^{\text{XP}}, r_{t+1}^{\text{XP}}, \boldsymbol{H}_{t+1}^{\text{XP}}, \text{ID}_1^{\text{XP}}, \text{ID}_2^{\text{XP}}\rangle$

17:        **if** $t \bmod t_{\text{update}} = 0$ **then**

18:            // Parameter Update

19:            $\mathcal{L}_{\Theta,\phi}(\mathcal{D}^{\text{SP}}, \mathcal{D}^{\text{XP}}) \leftarrow \textbf{COMPUTE\_LOSS}(\mathcal{D}^{\text{SP}}, \mathcal{D}^{\text{XP}}, \Theta, \phi, \bar{\phi})$

20:            **for** $\theta_i \in \Theta$ **do**

21:                $\theta_i \leftarrow \textbf{GRADIENT\_DESCENT}(\theta_i, \alpha, \nabla_{\theta_i}\mathcal{L}_{\Theta,\phi}(\mathcal{D}^{\text{SP}}, \mathcal{D}^{\text{XP}}))$

22:            **end for**

23:            $\phi \leftarrow \textbf{GRADIENT\_DESCENT}(\phi, \alpha, \nabla_{\theta_i}\mathcal{L}_{\Theta,\phi}(\mathcal{D}^{\text{SP}}, \mathcal{D}^{\text{XP}}))$

24:            $\bar{\phi} \leftarrow (1 - \bar{\alpha})\phi + \bar{\alpha}\phi$

25:            $\mathcal{D}^{\text{SP}} \leftarrow \{\}$

26:            $\mathcal{D}^{\text{XP}} \leftarrow \{\}$

27:        **end if**

28:        $t \leftarrow t + 1$

29:    **end while**

30: **end for**

31: **Return:** $\Theta$

---

environment are provided in Section B.1. Meanwhile, Section B.2 outlines the heuristics followed by the policies in $\Pi^{\text{eval}}$ for LBF, while Section B.3 outlines the heuristic followed by agents in the simple cooking environment.

### B.1 Cooperative Reaching

For Cooperative Reaching, we implement 11 heuristics as part of $\Pi^{\text{eval}}$. Each heuristic differs from others in terms of their way of selecting which reward-providing coordinates to move towards. Some heuristics also encourage teammates to follow the learner towards one of the existing reward-providing coordinates. The details of each heuristic used in Cooperative Reaching are provided below:

- **Heuristic H01.** This heuristic selects the action that gets a teammate closer to the closest reward-providing coordinate.

- **Heuristic H02.** This heuristic selects the action that gets a teammate closer to the furthest reward-providing coordinate from its initial position at the beginning of the episode.

- **Heuristic H03.** A teammate under this heuristic moves towards the closest optimal reward-providing coordinate.

- **Heuristic H04.** H4 moves an agent towards the furthest optimal reward-providing coordinate from a teammate's initial location in an episode.

- **Heuristic H05.** Same as H4, except that the learner only considers the suboptimal reward-providing coordinates instead of the optimal ones.

- **Heuristic H06.** Same as H5, except the teammate goes towards the closest suboptimal reward-providing coordinate.

- **Heuristic H07.** At the beginning of the episode, agents under this heuristic randomly select a reward-providing coordinate and move towards it.

- **Heuristic H08.** This heuristic moves a teammate towards the reward-providing coordinate closest to its counterpart agent's location.

- **Heuristic H09.** Same as H8, but only optimal reward-providing coordinates are considered as the teammate's destination.

- **Heuristic H10.** This heuristic moves the teammate towards its counterpart agent's location.

- **Heuristic H11.** This heuristic always randomly selects an action from the teammate's possible actions.

### B.2 Level-Based Foraging

Like Cooperative Reaching, we create diverse teammate heuristics requiring a learner to adapt their policies to achieve optimal collaboration. The ten heuristics used for LBF generally correspond to different ways of deciding the ordering to collect objects scattered in LBF's grid world. Details of each heuristic are provided below:

- **Heuristic H01.** The teammate attempts to collect whichever object is closest to its current location.

- **Heuristic H02.** At each timestep, the teammate computes the midpoint between the learner and its location. This teammate then attempts to collect whichever object is closest to this midpoint.

- **Heuristics H03-H08.** For heuristics H03 to H08, we assign a distinct random index from $\{1, 2, 3\}$ to each object at the beginning of each episode. Heuristics H03-H08 then collect the objects according to one of the 6 distinct possible orderings of the object index.

- **Heuristic H09.** The teammate always attempts to collect food closest to the learner's location.

- **Heuristic H10.** At the beginning of each episode, H10 identifies the object furthest from its location and attempts to collect it. Each time its target item is collected, H10 then attempts to collect the remaining object at the furthest distance from the current location of the controlled teammate.

### B.3 Simple Cooking

As the layout of our Simple Cooking is a ring, we consider two movement directions around the ring: clockwise and anti-clockwise. Each heuristic agent has a goal, such as "seek and process the nearest food." Once their goal has been completed, the heuristic agent finds a counter without any tools on it and stands on the empty space closest to the said counter.

- **Heuristic H1:** Seeks the nearest food in the clockwise direction, picks it up, and continues to move clockwise to the appropriate food processing counter, where it processes the food.

- **Heuristic H2:** Seeks the nearest food in the anti-clockwise direction, picks it up, and continues to move anti-clockwise to the appropriate food processing counter, where it processes the food.

- **Heuristic H3:** Takes the shortest path to the nearest food item, picks it up, and continues to take the shortest path to the appropriate food processing counter, where it processes the food.

- **Heuristic H4:** Seeks the furthest away food in the clockwise direction, picks it up, and continues to move clockwise to the appropriate food processing counter, where it processes the food.

- **Heuristic H5:** Seeks the furthest away food in the anti-clockwise direction, picks it up, and continues to move anti-clockwise to the appropriate food processing counter, where it processes the food.

- **Heuristic H6:** Same as Heuristic H3, except 25% of the time, the agent takes a uniform random action.

- **Heuristic H7:** This heuristic seeks the nearest processed food in the clockwise direction. It then picks up the processed food and checks whether the plate is on one of the counter or not. If the plate is on one of the counters, the agent moves clockwise to the plate in order to put the processed food. Otherwise, the agent goes towards the serving counter to place the food. Once there are no processed food to move, this agent moves clockwise to stand in front of an outer counter without any items on top.

- **Heuristic H8:** This heuristic is similar to H7 except that agents under this heuristic always move in an anti-clockwise direction.

- **Heuristic H9:** This heuristic is similar to heuristics H7 and H8 except that agents under this heuristic always decide its clockwise or anti-clockwise movement based on the shortest distance between its target object or location.

- **Heuristic 10:** This heuristic is similar to heuristics H7 except that agents under this heuristic will immediately put the processed food on the serving counter.

- **Heuristic H11:** This heuristic is similar to heuristics H8 except that agents under this heuristic always put the retrieved processed food on the service counter.

- **Heuristic H12:** This heuristic is similar to heuristics H9 except that agents under this heuristic will immediately put retrieved processed food on the service counter.

## C Experiment Hyperparameters

This section provides details of the hyperparameters and neural network architectures used in our teammate generation experiments.

- When optimising BRDiv, we run 32 parallel threads to collect self-play experiences during training. Meanwhile, the remaining methods use 160 parallel threads to gather self-play experiences used during their teammate generation process.

- Aside from the threads used to gather self-play experiences, we use 128 parallel environments to collect cross-play experiences when optimising BRDiv.

- All evaluated methods have their actor and critic networks updated every 8 timesteps.

- $\gamma$ is set to 0.99.

- The generated actor networks alongside the critic network are trained using Adam optimiser (Kingma & Ba, 2014) with a learning rate of $10^{-4}$.

- We clip the gradients of the model so that it always lies between -1 and 1.

- Each actor network corresponding to policies in the generated $\Pi^{\mathrm{train}}$ and $\mathrm{BR}(\Pi^{\mathrm{train}})$ are implemented as multilayer perceptrons. The size of these networks for each environment is detailed below:

    - **Cooperative Reaching.** The model comprises of four hidden layers with 128, 256, 256, and 128 neurons respectively.
    - **LBF.** The model comprises of two hidden layers, each consisting of 128 neurons.
    - **Simple Cooking.** Our network for this environment has two hidden layers with each layer having 256 neurons.

- We associated different weights to the optimised loss functions when generating $\Pi^{\mathrm{train}}$ using our proposed method, TrajeDi, Any-Play and the independent baseline. The weights of each loss function optimised by these methods are detailed below:

    - For all methods, the critic loss function for SP data is also set to 1.0. Meanwhile, BRDiv assigns a weight of 1.0 to the loss function that minimizes the critic loss function following cross-play interaction data.
    - For BRDiv, the weights of the losses optimised for training the actor networks is set to 25.
    - The Jensen-Shannon Divergence term maximised by TrajeDi is given a weight of $10^{-3}$. We arrive at this value after finding the largest possible weight from $\{10^{-1}, 10^{-2}, 10^{-3}, 10^{-4}\}$ that still ensures every policy in $\Pi^{\mathrm{train}}$ to achieve optimal performances in the environment when collaborating with its associated best response policy.
    - The weights of Any-Play's intrinsic reward to maximise diversity between populations is tuned in the same way as how we tuned the Jensen-Shannon Divergence weights for TrajeDi. This results in the intrinsic reward weights of $10^{-2}$, $10^{-3}$, and $10^{-3}$ for Cooperative Reaching, LBF, and Simple Cooking.
    - The classifier Any-Play uses to compute the intrinsic rewards uses the same architecture of other methods' critic networks. The term associated with the supervised learning loss utilised to train this classifier is also set to 1.
    - For TrajeDi, Any-Play, and the independent baseline, the weights associated with the term that maximises the self-play performance between a policy in $\Pi^{\mathrm{train}}$ and their associated best response policies is set to 1.

- We also use $\eta = 1$ for the polynomial weight weighting algorithm for PLASTIC Policy, which is the algorithm we use for our AHT experiments.

---

**Algorithm 2** Loss Computation

---

**Require:**

    Self-play and cross-play data, $\mathcal{D}^{\text{SP}}$ & $\mathcal{D}^{\text{XP}}$.

    Population actor network parameters, $\Theta$.

    Centralised critic parameters and target centralised critic parameters, $\phi$ & $\bar{\phi}$.

1:  **function COMPUTE_LOSS**$(\mathcal{D}^{\text{SP}}, \mathcal{D}^{\text{XP}}, \Theta, \phi, \bar{\phi})$

2:     $t_{\text{start}} \leftarrow$ first time in the buffers $\mathcal{D}^{\text{SP}}, \mathcal{D}^{\text{XP}}$

3:     $t_{\text{end}} \leftarrow$ latest time in the buffers $\mathcal{D}^{\text{SP}}, \mathcal{D}^{\text{XP}}$

4:     $V_{\text{target}} \leftarrow V(\boldsymbol{H}^{\text{SP}}_{t_{\text{end}}+1}, \text{ID}^{\text{SP}}, \text{ID}^{\text{SP}}; \bar{\phi})$

5:     $\mathcal{L}^{\text{SP}}_{\phi} \leftarrow 0$                                       ▷ Compute Self-Play Critic Loss

6:     **for** $t = t_{\text{end}}$ to $t_{\text{start}}$ **do**

7:         $V_{\text{pred}} \leftarrow V(\boldsymbol{H}^{\text{SP}}_t, \text{ID}^{\text{SP}}, \text{ID}^{\text{SP}}; \phi)$

8:         $V_{\text{target}} \leftarrow \begin{cases} r^{\text{SP}}_t, & \text{if episode terminates at } t \\ r^{\text{SP}}_t + \gamma V_{\text{target}}, & \text{otherwise.} \end{cases}$

9:         $\mathcal{L}^{\text{SP}}_{\phi} \leftarrow \mathcal{L}^{\text{SP}}_{\phi} + \frac{1}{2}(V_{\text{pred}} - V_{\text{target}})^2$

10:     **end for**

11:     $V_{\text{target}} \leftarrow V(\boldsymbol{H}^{\text{XP}}_{t_{\text{end}}+1}, \text{ID}^{\text{XP}}_1, \text{ID}^{\text{XP}}_2; \bar{\phi})$

12:     $\mathcal{L}^{\text{XP}}_{\phi} \leftarrow 0$                                      ▷ Compute Cross-Play Critic Loss

13:     **for** $t = t_{\text{end}}$ to $t_{\text{start}}$ **do**

14:         $V_{\text{pred}} \leftarrow V(\boldsymbol{H}^{\text{XP}}_t, \text{ID}^{\text{XP}}_1, \text{ID}^{\text{XP}}_2; \phi)$

15:         $V_{\text{target}} \leftarrow \begin{cases} r^{\text{XP}}_t, & \text{if episode terminates at } t \\ r^{\text{XP}}_t + \gamma V_{\text{target}}, & \text{otherwise.} \end{cases}$

16:         $\mathcal{L}^{\text{XP}}_{\phi} \leftarrow \mathcal{L}^{\text{XP}}_{\phi} + \frac{1}{2}(V_{\text{pred}} - V_{\text{target}})^2$

17:     **end for**

18:     $V_{\text{bootstrap}} \leftarrow V(\boldsymbol{H}^{\text{SP}}_{t_{\text{end}}+1}, \text{ID}^{\text{SP}}, \text{ID}^{\text{SP}}; \phi)$

19:     $\mathcal{L}^{\text{SP}}_{\Theta} \leftarrow 0$                                      ▷ Compute Self-Play Actor Loss

20:     **for** $t = t_{\text{end}}$ to $t_{\text{start}}$ **do**

21:         $M_{\text{baseline}} \leftarrow$ **TO_XP_MATRIX**$\big(\{V(\boldsymbol{H}^{\text{SP}}_t, i, j; \phi) | i, j \in 1, \dots, N\}\big)$

22:         $V_{\text{bootstrap}} \leftarrow \begin{cases} r^{\text{SP}}_t, & \text{if episode terminates at } t \\ r^{\text{SP}}_t + \gamma V_{\text{bootstrap}}, & \text{otherwise.} \end{cases}$

23:         $M_{\text{pred}} \leftarrow M_{\text{baseline}}$

24:         $M_{\text{pred},\text{ID}^{\text{SP}},\text{ID}^{\text{SP}}} \leftarrow V_{\text{bootstrap}}$         ▷ Replace matrix element of interacting populations

25:         $\mathcal{L}^{\text{SP}}_{\Theta} \leftarrow \mathcal{L}^{\text{SP}}_{\Theta} - \log(\pi(a^{1,\text{SP}}_t | H^{1,\text{SP}}_t; \theta_{\text{ID}^{\text{SP}}}) \pi^{\text{BR}}(a^{2,\text{SP}}_t | H^{2,\text{SP}}_t; \theta^{\text{BR}}_{\text{ID}^{\text{SP}}}))(\text{BRDiv}(M_{\text{pred}}) - \text{BRDiv}(M_{\text{baseline}}))$

26:     **end for**

27:     $V_{\text{bootstrap}} \leftarrow V(\boldsymbol{H}^{\text{XP}}_{t_{\text{end}}+1}, \text{ID}^{\text{XP}}_1, \text{ID}^{\text{XP}}_2; \phi)$

28:     $\mathcal{L}^{\text{XP}}_{\Theta} \leftarrow 0$                                     ▷ Compute Cross-Play Actor Loss

29:     **for** $t = t_{\text{end}}$ to $t_{\text{start}}$ **do**

30:         $M_{\text{baseline}} \leftarrow$ **TO_XP_MATRIX**$\big(\{V(\boldsymbol{H}^{\text{XP}}_t, i, j; \phi) | i, j \in 1, \dots, N\}\big)$

31:         $V_{\text{bootstrap}} \leftarrow \begin{cases} r^{\text{XP}}_t, & \text{if episode terminates at } t \\ r^{\text{XP}}_t + \gamma V_{\text{bootstrap}}, & \text{otherwise.} \end{cases}$

32:         $M_{\text{pred}} \leftarrow M_{\text{baseline}}$

33:         $M_{\text{pred},\text{ID}^{\text{XP}}_1,\text{ID}^{\text{XP}}_2} \leftarrow V_{\text{bootstrap}}$         ▷ Replace matrix element of interacting populations

34:         $\mathcal{L}^{\text{XP}}_{\Theta} \leftarrow \mathcal{L}^{\text{XP}}_{\Theta} - \log(\pi(a^{1,\text{XP}}_t | H^{1,\text{XP}}_t; \theta_{\text{ID}^{\text{XP}}_1}) \pi^{\text{BR}}(a^{2,\text{XP}}_t | H^{2,\text{XP}}_t; \theta^{\text{BR}}_{\text{ID}^{\text{XP}}_2}))(\text{BRDiv}(M_{\text{pred}}) - \text{BDiv}(M_{\text{baseline}}))$

35:     **end for**

36:     **Return:** $\mathcal{L}^{\text{SP}}_{\phi} + \mathcal{L}^{\text{XP}}_{\phi} + \mathcal{L}^{\text{SP}}_{\Theta} + \mathcal{L}^{\text{XP}}_{\Theta}$

37: **end function**

---

