# OpenReview forum: "Generating Teammates for Training Robust Ad Hoc Teamwork Agents via Best-Response Diversity"
_TMLR — Accepted by TMLR_

### Review · Reviewer_XuvG · 2023-03-04

**Summary Of Contributions:**

This paper proposes a methodology for generating diverse teammates so as to obtain improved robustness in the resulting learning when collaborating with new teammates. Diversity is obtained by optimising the Best-Response Diversity (BRDiv) metric, which is also introduced in this paper. The authors evaluate their approach in a number of environments with varying coordination strategies.

**Audience:**

Yes

**Broader Impact Concerns:**

No concerns

**Claims And Evidence:**

Yes

**Requested Changes:**

# Comments / suggestions / questions
1. Throughout the paper you refer to the "best-response policy" but it's not clear what you mean by that, nor how it is learned.
1. Throughout the paper you use the term "superficial differences", but it's not clear to me what you mean by that. For example, in Figure 1 you say (a) and (b) only have superficial differences, but they seem quite different to me.
1. In the caption of Figure 1 you say "an AHT learner will only acquire the skill to pass...". This suggests this is the case for _all_ learners of this type, which I doubt is true. Also, why is learning the passing skill bad? Same issue in the main text in the introduction when you say "A learner trained to collaborate with these policies will only acquire expertise in passing...".
1. I'd suggest revisiting Figure 1 and the example provided. It's not clear what message you're trying to convey.
1. In the introduction you say "they do not encourage improved learner robustness", please be more specific as to what kind of robustness you're referring to.
1. The second to last sentence in page 2 is hard to parse. "teammate policies" appears twice in the sentence but it's not clear they are referring to the same thing. Please expand and clarify more.
1. In the top of page 3, it's not clear what you mean by "forcing". It suggests you're artificially making things better for your method?
1. In the first sentence of section 3.2, how does $K$ relate to $N$?
1. In the second paragraph of section 3.2, is the "position" of $\pi^i$ (in terms of action selection) is fixed throughout training? Order is important, so stating that assumption would be best.
1. In equation (1), I assume the size of $\mathbf{\pi^{-i}}$ is $N-1$?
1. In the last paragraph of page 4, when you say "feasible teammate policies", are these evaluated with the learner $\pi^i$ or not? If yes, how ($\pi^i$ hasn't been learned as far as I understand)? If not, what is used instead? Just a regular learner? Please clarify.
1. In the first sentence of section 4.1 you say "a suitable $\Pi^{train}$ for AHT...". It's not clear what this means, nor why we want to avoid it. It's phrased weirdly, so it's hard to parse. The equations that come after do clarify things, so lean on that when explaining.
1. In the second equation in section 4.1, is $a_t$ the combination of $a^1_t$ and $a^2_t$?
1. In the last paragraph of section 4.1, when you say "by the existence of a common...", I'd say your method is in a sense more superficial since it's only comparing based on returns. Thus, two very different policies, but with the same returns, will be classified as "similar", when they probably shouldn't.
1. In the first paragraph of section 4.2, when you say "Extending our proposed diversity metric...", I'm not sure about this, in particular with respect to the ordering assumption I mentioned above.
1. In equation (4) you switch from $j,k$ to $1,2$.
1. Isn't equation (4) just the same as the second equation in section 4.1?
1. The last paragraph in page 6 could benefit from a figure, as there's a fair bit of notation, but it should be reasonable to make a figure to clarify.
1. In **Data Collection** in seciton 5, when you say "First, we collect self-play experiences...", these are changing throughout training, right? It's a little confusing as it's worded here as if they're fixed and simply being chosen from a pre-existing set. See also my comment on algorithm below.
1. In **Action and Centralised Critic Architecture** in section 5, when you say "the trained actors", these are already trained, or you're training them? The wording is a little confusing.
1. In equation (9) it's not clear how you arrived at this loss function from the preceding discussion.
1. Where are the $C^{pred}$s from equation 10 used? It seems they should be used in equation (9) (from the wording), but they're not there.
1. In **Level-based Foraging** of section 6.1, when you say "the total levels of agents...", are the levels added when both agents are in the same cell?
1. In the first paragraph of page 10, just to confirm: agents can both be in the same position?
1. In the third paragraph of section 6.2, just to confirm: Policies in $\Pi^{train}$ are fixed at this point, yes?
1. In the fourth paragraph of section 6.2, make it clear that you also evaluate with policies generated by other _methods_, not just different seeds. This becomes clearer later, but you could clarify nmore at this point as well.
1. In the last paragraph of section 6.2 when you say "with each unknown teammate policy type from $\Pi^{train}$" I think you mean $\Pi^{eval}$?
1. In the second-to-last paragraph of section 6.3, when you say "a classifier that identifies the population... based on an observed state and its action", given that state and action spaces used are very small, is a state-action pair really enough to classify teammates? Given that this classifier is central to AnyPlay, it seems like it places it at a pretty stron disadvantage.
1. In **Ablations of BRDiv** of section 6.3, when you say "Furthermore, when computing MAA2C's actor loss...", this is still for Independent baseline?
1. The grids in Figures 4 and 5 are kind of hard to interpret. You should consider using something like interquantile mean (see [RLiable](https://github.com/google-research/rliable) and [paper](https://proceedings.neurips.cc/paper/2021/hash/f514cec81cb148559cf475e7426eed5e-Abstract.html)) to provide a more digestible aggregate summary. It seems scores are normalized, so it should be relatively easy to add with rliable.
1. In addition to the last point, a useful plot to generate would be a scatter plot of BRDiv score against the final agent performance. This can allow us to determine if there _is_ a correlation between them, and it would further justify your choice of this metric.
1. Is Figure 7 (b) supposed to be better than 6(b)? It's not clear that this is the case.
1. In the bottom of page 15, when you say "Learning from such highly optimal policies makes...", why was this not an issue with other environments?
1. In the last sentence of section 6.5 (right before section 7) you say "a specific return threshold at the given task". What would be a good threshold here?
1. In Algorithm 1, where the "best response policies" chosen?




# Nits
1. In the first sentence of section 3, remove the last 's' from "formalises".
1. In the caption of Figure 2 you say "a shared critic network (green box)", but I think this should be blue box.
1. In the last line of page 8 you wrote "chapter", but you mean "section".

**Strengths And Weaknesses:**

# Strengths
The authors are tackling an interesting, and challenging, problem that does not receive much attention, but will likely start receiving more as the community develops more capable agents.
The method proposed by the authors makes sense, and I am reasonably convinced it generates diverse policies that benefit the learning agent. The authors performed a large set of experiments that help in showcasing the advantages of their proposed method.

# Weaknesses
My main concern with this paper is in its clarity. There are many points (detailed below) where the writing could be improved.
I also have some concerns with the way the baselines were run, which are detailed below.

---

> ### Author Response · Authors · 2023-03-17
> **Response to Reviewer XuvG (Part 1)**
>
> We thank reviewer XuvG for their comments, suggestions, and questions, which helped us improve the clarity of our work.
>
> (1) Given a policy $\pi$, a best-response policy ($\pi^{*}$) to $\pi$ is the policy that achieves the highest expected returns when collaborating with a teammate following policy $\pi$. The best-response policy to $\pi$ can be estimated by training a learner's policy to maximize the team returns when interacting with teammates with policy $\pi$. This is why we maximize the trace of the cross-play matrix in the BRDiv metric on Equation 6 (i.e. to ensure we correctly estimate the best-response policy to each generated teammate policy). We will explain what BR policies are in the introduction. A more formal definition will be added in Section 3.
>
> (2) Superficial differences refer to differences among generated teammate policies for which our learning agent has the same best-response policy. Such differences are superficial in AHT since they do not encourage a learner to discover different best response policies for effective collaboration, which we illustrate in Figures 6a and 7a as important for robust collaboration (i.e. producing high returns) with previously unseen teammate policies. In the revised document, we will also add a Figure based on Figures 6a & 7a in our introduction to illustrate the importance of discovering policies with different best responses to improve robustness. Since it is also related to Questions 3 and 4, we will explain why superficial differences do not necessarily translate to improved robustness in our answers to Questions 3 and 4.
>
> (3 & 4) Figure 1 intends to illustrate how superficial differences in generated teammate policies contribute less towards improved AHT learner robustness. Superficial differences occur when we generate different teammate policies that share the same best response policy. When an AHT learner is trained by optimizing Equation 2 against $\Pi^{\text{train}}$ having superficial differences, the learner only learns the shared best response to the superficially different generated policies.
>
> Superficial differences may cause an AHT learner to produce low returns against certain unknown teammate types (i.e. not being robust) if multiple best-response policies to solve an environment exist. Figure 6(a) illustrates an example of superficial differences in an environment with 4 different best response strategies (i.e. corresponding to moving towards different corners in the environment). When trained against $\Pi^{\text{train}}$ in Figure 6a to maximize Equation 2, the learner will only learn a best-response policy that either moves towards the top-right or bottom-left corner depending on the movement of its teammate. Against a teammate moving towards the bottom-right or top-left corner, the learner will, by contrast, produce low returns. This is because the learner will not learn a policy to move to the bottom-right or top-left corner, which both are not the best responses to any policies in $\Pi^{\text{train}}$ illustrated by Figure 6a.
>
> Contrast the superficially different teammate policies in Figure 6a with Figure 7a, where $\Pi^{\text{train}}$ go towards each reward-generating corner. When the AHT learner maximizes Equation 2 against $\Pi^{\text{train}}$ in Figure 7a, it will learn to follow teammates to any reward-generating corner to best respond to  $\Pi^{\text{train}}$ containing teammate policies moving towards all four corners. This makes the AHT learner trained with  $\Pi^{\text{train}}$ from Figure 7a achieve high returns against a broader range of teammate policies (i.e. more robust) compared to a learner trained with teammates illustrated in Figure 6a.
>
> We will replace Figure 1 with a more straightforward and clearer example based on an environment similar to the one producing Figures 6a & 7a. Also, we will describe in more detail why superficial differences cause teammates to produce low returns against specific types in the introduction.
>
> (5) To clarify, our robustness measure is defined in Equation 1. We will also specify our robustness measure in the introduction section of the revised paper version.
>
> (6) To clarify, these terminologies refer to the policies involved in the cross-play matrix computation from Section 4.2 Equation 5:
>
> Generated teammate policies →  $\Pi^{\text{train}}$
>
> Best-response policies for other generated teammate policies. → $\text{BR}(\Pi^{\text{train}}) = \\{ \pi^{\*} |  \exists(\pi\in \Pi^{\text{train}}), \pi^{*} = \text{argmax}_{\pi^{\text{learner}}}  \mathbb{E}\_{a^{i}\_{t}\sim\pi^{\text{learner}}, a^{-i}\_{t}\sim\pi} \left[ \sum\_{t=0}^{\infty} R\_{t}(s\_{t},a\_{t}) \right]\\} $
>
> We will attempt to explain this idea better in the introduction.
>
> (7) We will replace this word with “optimize” to indicate how we introduce a different objective function to produce more desirable $\Pi^{\text{train}}$.

---

> ### Author Response · Authors · 2023-03-17
> **Response to Reviewer XuvG (Part 2)**
>
> (8) N is the number of agents in the environment. K is the number of policies we generate. We will make this clearer by adding a sentence that defines both.
>
> (9) To clarify, we assume that i is fixed (i.e. the learner’s id remains the same) throughout learning and evaluation. Furthermore, the agents (learner, i, and its teammates, -i) choose their actions at the same time at every timestep without having any ordering between them (as in any Dec-POMDPs).
>
> (10)  Correct. $\pi^{-i}$ jointly select the actions of the N-1 agents in the environment. We will try to make this clearer.
>
> (11) To clarify, we add that sentence to summarize the assumption in the seminal AHT paper by Stone et al. (2010) regarding  $\Pi^{\text{eval}}$. In that work, any policy $\pi$ can be a member of $\Pi^{\text{eval}}$ as long as there exists a policy $\pi^{‘}$ that achieves expected returns above an expert-defined threshold when collaborating with $\pi$.
>
> In more recent works in AHT (Rahman et al., 2021; Papoudakis et al., 2021; Gu et al., 2023), $\Pi^{\text{eval}}$ can either consist of heuristic-based or RL-based policies. Note that the use of heuristics or RL-based policies does not conflict with the assumption of Stone et al. (2010) since the selection threshold can always be lowered to allow any heuristic or policy to get included in $\Pi^{\text{eval}}$. As highlighted in Section 6.2, $\Pi^{\text{eval}}$ in our work consists of heuristic-based or RL-based policies trained via various teammate generation methods.
>
> (12) We will revise the sentence to better define the notion of suitability. In general,  $\Pi^{\text{train}}$ is deemed more suitable if it does not exhibit superficial differences. The first expression of Section 4.1 is merely a formalization of superficial differences, which we try to avoid. We will also link this to our explanation of why we avoid superficial differences (answer to Q3&4 above).
>
> (13) Yes, that is correct. We will highlight this more clearly in the paper.
>
> (14) We thank the reviewer for pointing this out. To clarify, we have defined “superficial” in terms of whether those differences help improve the learner's robustness (i.e. returns achieved) against unknown teammate policies. Following BRDiv’s higher returns in our experiments and reasoning behind avoiding superficial differences (answer to Q3&4), we disagree that policies that share the same best responses are less superficial (according to our definition of whether it induces more robust agents) to our generated policies.
>
> Again, we reiterate that generating $\Pi^{\text{train}}$ with policies having distinct best-response policies is crucial due to the learning objective of AHT training algorithms, which is to maximize Equation 2. Under this learning framework, existing AHT learning methods will essentially try to model teammates encountered during training and approximate the best response to their policies. If somehow $\Pi^{\text{eval}}$ contains policies that have different best responses to those in $\Pi^{\text{train}}$, then the learner cannot be expected to produce high returns against them (i.e. the learner will appear less robust during evaluation). That is why we encourage the creation of  $\Pi^{\text{train}}$ containing policies with as many different best responses to encourage the learner to be competent at executing a wider range of best response policies during evaluation. This hopefully translates to a higher likelihood of achieving high returns when facing unknown teammate policies from $\Pi^{\text{eval}}$.
>
> (15) Concerning our answer to Q9, as long as (i) the learner’s id is fixed and (ii) all agents choose their actions jointly, extending BRDiv to more than two agents is straightforward. Specifically, we must jointly train multiple policies to control each agent in -i (the set of agents other than the learner).
>
> (16) Thank you for noticing this. We will switch 1,2 to j,k in the revision.
>
> (17)  Both equations indeed denote expected returns. It’s just that Equation 4 is the expected returns conditioned on agents’ previous experiences $H$.
>
> (18) Figure 2 (blue table and the arrows pointing towards it) already conveys the message we want to impart via that paragraph. Nonetheless, we will mention parts of Figure 2 that are relevant to that paragraph in the revision.
>
> (19) That is correct. Similar to each update for MAA2C, the collected data is gathered, used for a single update, and discarded afterwards. We will clarify this in the revised version.
>
> (20) We train these actor networks during the teammate generation process. At the end of the teammate generation process, these actors will be the generated teammate policies. We will clarify this in the revision.

---

> ### Author Response · Authors · 2023-03-17
> **Response to Reviewer XuvG (Part 3)**
>
> (21) This Equation is based on the MAA2C update, where we try to maximize the product of agents’ joint action log-likelihood and an advantage function. Since the generated policy and the BR policy choose their actions independently, the joint action probability becomes a product of individual agents’ action likelihood (first line in Equation 9 inside the logarithm). Meanwhile, our advantage function differs from the usual advantage function in MAA2C to factor in how we use the BRDiv metric to compute this function (Equation 8). We will clarify this in the revision.
>
> (22) It is used in Equation 8 when defining the BRDiv-based advantage function. We decided to introduce $C^{pred}$ without defining it first because we believe it is easier to explain the intuition behind the advantage function first (Equation 8) before detailing its computations. We will mention in the revisions that the terms defined in Equation 10 are used to evaluate Equation 8.
>
> (23) Agents cannot occupy the same locations in LBF. Agent levels are added only if they jointly select the collection action from different locations adjacent to an object. For instance, this happens if agents in (0,1) and (1,0) decide to collect an object in (1,1). We will clarify this in the revisions.
>
> (24) No, agents cannot occupy the same space. We now realise that we had not mentioned this. This will be clarified in the revision.
>
> (25) Correct. $\Pi^{\text{train}}$ remains fixed after the first stage (i.e. teammate generation) of evaluation.
>
> (26) We agree with this suggestion.  We will add this to the revision.
>
> (27) Thank you for noticing this typing error from us. We will fix this in the revision.
>
> (28) We argue that AnyPlay is not disadvantaged compared to the proposed method. Since we use fully observable environments for evaluation, we also compute our diversity metric (BRDiv) based on the state of the environment, like AnyPlay. This ensures fairness when comparing both methods. We can clarify this in the revision.
>
> (29) Correct. We specifically choose MAA2C for generating teammates under the Independent baseline.
>
> (30) We intend to keep the per-type performance breakdown since it provides evidence of a few important points in our discussion (i.e. which types of teammates do BRDiv succeed/fail to collaborate against, etc.). Nevertheless, we will explore the possibility of changing the reported measures to the metrics proposed in the RLiable paper. On another note, we can add another Figure that aggregates the evaluation results from different policy types into a single visualization and add it to the appendix.
>
> (31) Thank you for suggesting this. We can try adding this suggested scatterplot in the revisions.
>
> (32) Yes. Please refer to our answer to Q3&4 for why 6a is not as good as 7a.
>
> (33) We believe it results from our experiments' environment episode lengths. The way Cooperative Reaching and LBF end in 20, and 50 timesteps means that highly suboptimal policies (i.e. one that spans more than 20/50 timesteps) will never be discovered in these environments. Such suboptimal policies will not be able to maximize the maximized cross-play matrix trace in Equation 9 since the environment will end before these policies solve the task.
>
> (34) To our best knowledge, there’s no work specifying a general framework to set this threshold. As highlighted in answer to Question 11, this threshold is typically used to formalise suboptimal teammates' existence in the environment. This threshold also does not play any role in solving AHT problems.
>
> (35) In Algorithm 1, the best response policy for a generated policy should be the one interacting with it during self-play training. This can be seen in Line 10, where a generated policy (i.e. the one where $a^{1,\text{SP}}\_{t}$ is sampled from) interacts with the best response policy (i.e. the one where $a^{2,\text{SP}}\_{t}$ is sampled from).
> Upon revisiting Algorithm 1, we realized how to better highlight the difference between a generated policy and its best response by assigning different symbols to them. We will add this change to our revision.
>
> Cited Works:
>
> Peter Stone, Gal A. Kaminka, Sarit Kraus, and Jeffrey S. Rosenschein. Ad hoc autonomous agent teams: Collaboration without pre-coordination. AAAI 2010.
>
> Arrasy Rahman, Niklas Höpner, Filippos Christianos, and Stefano V. Albrecht. Towards open ad hoc teamwork using graph-based policy learning. ICML 2021.
>
> Georgios Papoudakis, Filippos Christianos, and Stefano Albrecht. Agent modelling under partial observability for deep reinforcement learning. NeurIPS 2021.
>
> Pengjie Gu, Mengchen Zhao, Jianye Hao, Bo An. Online Ad Hoc Teamwork under Partial Observability. ICLR 2022.

---

### Review · Reviewer_8v61 · 2023-03-07

**Summary Of Contributions:**

The paper deals with the Ad-Hoc teamwork framework, where a learner must interact and collaborate with previously unencountered teammates, without any prior coordination. Existing work already addresses this challenge using methods that train the learner given a diverse set of teammate policies (e.g. handcrafted ones tailored manually for the demain).

However, such a solution is not scalable, as such a manual intervention is required for each new domain to be captured. One possible solution is applying information theoretic metrics of diversity when generating teammates to train with, but such a simplistic metric can result in teammates with superficially different behaviors that are in reality quite similar, and not adding much robustness to the learner.
The authors propose a teammate generation algorithm which optimizes for a diversity metric called BRDiv, best-response diversity. This metric measures diversity based on the compatibility between teammate policies based on the joint returns.

The authors also provide an empirical evaluation in an environment where multiple valid coordination strategies are present, and contrast them with baselines based on approaches that optimize for information theoretic diversity (and also with the case where no optimization for diversity is done). The results show that the approach results in a more meaningfully diverse set of teammates to train with, hence improving the learners performance and robustness.

**Audience:**

Yes

**Broader Impact Concerns:**

I spotted no broader impact concerts.


**Claims And Evidence:**

Yes

**Requested Changes:**

In terms of presentation, I do feel like the authors could have done a better job at contrasting their approach with existing methods in a better way. A table comparing the various approaches would have been very useful (e.g. what the metric measures, how it is used, the training pipeline etc.). In particular, MAA2C based optimization with a metric, which is a key building block, should receive more attention.

Also, figures 4+5 is very busy, and perhaps there is an easier way to convey that message?

In terms of the empirical analysis, I think the environments presented here already give enough evidence of the value of the approach. However, including more settings and more difficult tasks could help show whether this principle is generally applicable. For instance, you have one environment that is akin to overcooked, but quite a simplistic one. A more difficult environment would be very useful. Possible games with many more agents? Could further multiagent AI gyms be used here?


**Strengths And Weaknesses:**

Overall, I really love the topic of ad-hoc teamwork, and I think that the key motivation of having meaningfully diverse policies to train with is very appealing.

The paper gives an overview of the topic of ad-hoc teamwork from its early days, which I find very helpful. Similarly, the description of the method is quite clear (with the technical figures adding a lot of value). I think the relation and differences with existing approaches could be presented in more depth (see below). Some of the figures are a bit cumbersome (see below), or could be simplified.

The empirical analysis is quite convincing but limited in terms of coverage and the simplicity of the environments (see suggestions below - mostly adding more environments, and having slightly more complex ones to show that the approach is not tailored to few specific domains).

---

> ### Author Response · Authors · 2023-03-17
> **Response to Reviewer 8v61**
>
> We first thank reviewer 8v61 for their insightful comments on our work.
>
> **Contrasting Proposed Approach & Baselines**
> > In terms of presentation, I do feel like the authors could have done a better job at contrasting their approach with existing methods in a better way. A table comparing the various approaches would have been very useful (e.g. what the metric measures, how it is used, the training pipeline etc.). In particular, MAA2C based optimization with a metric, which is a key building block, should receive more attention.
>
> Thank you for this suggestion. We will include the following table in Section 6.3 to help readers identify the difference between evaluated methods.
>
> | Diversity Metric \ Characteristics |                       Optimization Method                      | Self-Play Data | Cross-Play Data | Policy ID Classifier |
> |------------------------------------|:--------------------------------------------------------------:|:--------------:|:---------------:|:--------------------:|
> | BRDiv                              |                        MAA2C (Eq 7 & 9)                        |       Yes      |       Yes       |          No          |
> | Independent                        |                        MAA2C (Eq 7 & 9)                        |       Yes      |        No       |          No          |
> | TrajeDi                            | MAA2C (Eq 7 & 9) With Jensen Shannon Divergence Auxiliary Loss |       Yes      |        No       |          No          |
> | Any-Play                           |    MAA2C (Eq 7 & 9) With Classifier-based Intrinsic Rewards    |       Yes      |        No       |          Yes         |
>
> Although we can add the expressions for JSD and the AnyPlay intrinsic reward to the table, we prefer to omit this from the table since it’ll make the table rather cluttered. As an alternative, we will point to Equations from the works of Lupu et al. (2021) and Lucas et al. (2022) that evaluate the auxiliary loss/intrinsic rewards.
>
> **Cluttered Figures**
> > Also, figures 4+5 is very busy, and perhaps there is an easier way to convey that message?
>
> Similar to our answer to Question 30 from reviewer XuvG, we plan to keep the per-type performance breakdown as it provides insights regarding the types of teammates our method can (or cannot) collaborate with. These insights later provide important discussion points for our analysis section regarding our method’s strengths and weaknesses. However, we will provide an accompanying figure that better aggregates the evaluation results across different teammate types.
>
> **Experiment Environments**
> > In terms of the empirical analysis, I think the environments presented here already give enough evidence of the value of the approach. However, including more settings and more difficult tasks could help show whether this principle is generally applicable. For instance, you have one environment that is akin to overcooked, but quite a simplistic one. A more difficult environment would be very useful.
>
> We also agree with the reviewer that the three environments used in our experiments already provide sufficient evidence of the effectiveness of our proposed method. For overcooked, we believe that our Simple Cooking environment is at least as complex as existing Overcooked-based RL environments (Wang et al., 2020; Yu et al., 2023) because of the many subtasks that have to be completed in our environment. While some existing Overcooked environments only need to complete 2-4 subtasks to finish a recipe, our Simple Cooking environment requires 7 subtasks to be completed to finish the recipe (i.e. <i> Bring tomato to chopping counter, <ii> Bring carrot to the blending counter, <iii>Chop tomatoes, <iv> Blend carrots, <v> Put the chopped tomato on a plate, <vi> Put blended carrot on a plate, <vii> deliver the plate to serving counter). Furthermore, the constricted hallway that restricts the movement of agents also increases the difficulty for agents in terms of navigating through the environment. This already results in around 100 million timesteps to solve the Simple Cooking environment.
>
> **Games With More Agents**
> > Possible games with many more agents? Could further multiagent AI gyms be used here?
>
> In our current version of the method, we focused on two-player games. In future work, we look forward to expanding to more players. We believe our current method is general enough and can be used with any cooperative two-player multi-agent gym environment. We plan to make our code available to the public, including the environment, to allow the replication of our results.
>
> Cited Works:
>
> Sarah Wu, Rose E Wang, James Evans, Joshua Tenenbaum, David Parkes, Max Kleiman-Weiner. Too Many Cooks: Coordinating Multi-agent Collaboration Through Inverse Planning. AAMAS 2020.
>
> Chao Yu, Jiaxuan Gao, Weilin Liu, Botian Xu, Hao Tang, Jiaqi Yang, Yu Wang, Yi Wu. Learning Zero-Shot Cooperation with Humans, Assuming Humans Are Biased. ICLR 2023.

---

### Review · Reviewer_GwjL · 2023-03-12

**Summary Of Contributions:**

The paper presents a method for automated teammate policy generation in ad hoc teamwork scenarios.
Recent approaches have attempted to improve the robustness of the learner by optimising information-theoretic diversity metrics to generate teammate policies, however resulting in lack of useful diversity due to naive objectives (generating diversity with "superficial differences").

In contrast, the proposed method optimizes the Best-Response Diversity (BRDiv) metric, which measures diversity using reward distributions across a population of training agents, setting the objective to minmax certain teammate policies vs others in the population.

BRDiv is tested in a few simple environments, demonstrating that learn ad hoc agents are better at dealing with a set of unseen teammate policies, as well as being qualitatively diverse.


**Audience:**

Yes

**Claims And Evidence:**

Yes

**Requested Changes:**

Overall, I enjoyed the manuscript, and I don't have any strong change requirements. However:

1. I would *love* to understand the reason behind the lack of reutilization of previous ad-hoc environments (even if only some of the ones contained in the literature presented within the manuscript), or even some common MARL ones, since it feels like it should be generally possible to modify them to be compliant to the AHT setting.

2. I would like to see a better intuitive explanation for why it is necessary and worthwhile to minimize this particular interpretation of "superficial differences".


**Strengths And Weaknesses:**

 I really enjoyed reading this paper.

# Strengths

1. The manuscript is well written. It lays down enough background such that an expert reader outside of the standard ad-hoc MARL literature (such as myself) can appreciate the imposed constraints over BRDiv, and the various architectural decisions.

2. The BRDiv method is clever, is ultimately a simple objective that can be applied to a large set of MARL algorithms, and seems to be well justified from both intuition-wise as well as from a theoretical perspective.

3. The authors have gone above and beyond in evaluating BRDiv in the chosen environments, and I really commend them in how clear Section 6 is. I wish RL papers were generally setup in such manner.

# Weaknesses

1. I found it puzzling that the authors chose to implement their own toy environments. According to the background section, there exists plenty of literature on ad hoc learning, and that includes many environments. Considering that the manuscript essentially argues for improving a particular kind of diversity (i.e. reducing "superficial differences" in policy space), it would have been good to see this an improvement in this diversity metric translating into an improvement over common ad hoc environments.

2. The new diversity metric / requirement is a key concept in this paper, but its presentation is a bit lacking. It is quickly introduced from a high level perspective in Section 1, in which Figure 1 does most of the heavy lifting. However, the manuscript afterward tends to assume its importance without necessarily arguing for it (outside of the positive empirical results). In particular, the "superficiality" in policy space is quite narrowly defined, and seems to be highly dependent on the environment (or even subjective!). It feels like a broader discussion and/or presentation might be necessary here.

---

> ### Author Response · Authors · 2023-03-17
> **Response to Reviewer GwjL**
>
> We first thank reviewer GwjL for their constructive comments.
>
> **Environment Choice**
> > I found it puzzling that the authors chose to implement their own toy environments. According to the background section, there exists plenty of literature on ad hoc learning, and that includes many environments. Considering that the manuscript essentially argues for improving a particular kind of diversity (i.e. reducing "superficial differences" in policy space), it would have been good to see this an improvement in this diversity metric translating into an improvement over common ad hoc environments.
>
> > I would love to understand the reason behind the lack of reutilization of previous ad-hoc environments (even if only some of the ones contained in the literature presented within the manuscript), or even some common MARL ones, since it feels like it should be generally possible to modify them to be compliant to the AHT setting.
>
> When deciding which environments to use, we consider whether the environment has multiple cooperation strategies that do not share the same best response policy for optimal collaboration, which means the learner really needs to adapt to the behaviour of the other agent. As highlighted in Section 6.5, all three environments in our work fulfil this criterion as many different teammate strategies can solve the underlying collaboration task. In Cooperative Reaching, multiple strategies correspond to the distinct locations teammates can move towards to get rewarded. The different strategies in LBF correspond to different orderings followed by teammates when collecting the scattered objects. Simple Cooking environment also has different subtask allocation and completion strategies for effective collaboration.
>
> Regarding task complexity and the number of strategies to discover, we would argue that our simple cooking environment is more complex than some existing Overcooked environments (Wu et al., 2020; Yu et al., 2023) where there are only 2-4 subtasks to finish a recipe. Compare this to our Simple Cooking environment where 7 subtasks have to be done to finish the recipe (i.e. <i> Bring tomato to chopping counter, <ii> Bring carrot to the blending counter, <iii>Chop tomatoes, <iv> Blend carrots, <v> Put the chopped tomato on a plate, <vi> Put blended carrot on a plate, <vii> deliver the plate to serving counter). Furthermore, the constricted hallway that restricts the movement of agents also increases challenges for agents in navigating through the environment.
>
> On another note, we also use an environment commonly used for AHT. LBF is a commonly used environment for MARL and AHT evaluation (Papoudakis et al. 2021; Mirsky et al. 2022).
>
> **Intuition Behind Proposed Method**
> > The new diversity metric / requirement is a key concept in this paper, but its presentation is a bit lacking. It is quickly introduced from a high level perspective in Section 1, in which Figure 1 does most of the heavy lifting. However, the manuscript afterward tends to assume its importance without necessarily arguing for it (outside of the positive empirical results). In particular, the "superficiality" in policy space is quite narrowly defined, and seems to be highly
>
> > I would like to see a better intuitive explanation for why it is necessary and worthwhile to minimize this particular interpretation of "superficial differences".
>
> We agree with the reviewer that the motivation to minimize superficial differences is not entirely clear in the current version of our manuscript. This was also a concern raised by reviewer XuvG. We provide extended reasoning behind our methods’ importance to improve learner robustness for AHT in our answer to Questions 3 & 4 from reviewer XuvG. Please see our comments there. In our revised manuscript, we will modify the introduction to include this clarified explanation and a new figure to convey our proposed method's motivation better.
>
> Cited Works:
>
> Sarah Wu, Rose E Wang, James Evans, Joshua Tenenbaum, David Parkes, Max Kleiman-Weiner. Too Many Cooks: Coordinating Multi-agent Collaboration Through Inverse Planning. AAMAS 2020.
>
> Chao Yu, Jiaxuan Gao, Weilin Liu, Botian Xu, Hao Tang, Jiaqi Yang, Yu Wang, Yi Wu. Learning Zero-Shot Cooperation with Humans, Assuming Humans Are Biased. ICLR 2023.
>
> Georgios Papoudakis, Filippos Christianos, Lukas Schäfer, Stefano V Albrecht. Benchmarking Multi-Agent Deep Reinforcement Learning Algorithms in Cooperative Tasks. NeurIPS 2021.
>
> Reuth Mirsky, Ignacio Carlucho, Arrasy Rahman, Elliot Fosong, William Macke, Mohan Sridharan, Peter Stone, Stefano V Albrecht. A survey of ad hoc teamwork: Definitions, methods, and open problems. EUMAS 2022.

---

### Author Response · Authors · 2023-03-22
**Updates to the paper following the provided reviews**

We first thank all reviewers for their highly constructive feedback on the original version of our paper.

Following the reviewers' feedback, we have uploaded a revised version of our paper. In this latest version, we have made the following changes to the document:

**Major Modifications**

- Modifications that aim to explain better the rationale behind improving agent robustness through reducing superficial differences between generated policies. Parts of these modifications are located in the following locations:

    a) Introduction (Section 1).

    b) Example contrasting the consequences of generated teammates with high and low superficial differences (Figure 1).

    c) Description of desirable diversity for AHT (Section 4.1).

- Improvements regarding descriptions that contrast the proposed approach with baselines. Modifications related to this are provided in:

    a) Tabular summary of the difference between the proposed approach and baselines across multiple categories (Table 1).

    b) Description of the different loss functions or training processes behind baseline approaches (Section 6.3).

- Adding environment selection criteria in Section 6.1

- Additional visualisations on aggregated return statistics (i.e. interquartile mean <IQM> of returns) resulting from different evaluation scenarios. Modifications related to this are provided in:

    a) Updated experiment protocol in terms of reporting IQM (Section 6.2)

    b) Reported IQM of returns (Figure 4)

**Minor Modifications:**

- Adding extra information related to various clarification points from reviewers.

- Fixing typing errors related to symbols.

- Improving the clarity of a few sentences.

---

### Decision · Action_Editors · 2023-04-26

**Recommendation:** Accept as is

**Comment:**

This paper presents a method for learning policies for ad-hoc team formation,  i.e., situations with new teammates without pre-existing coordination strategies. Previous work has achieved that by placing, at training time, agents into situations where they are asked to interact with as much diverse teammates as possible. As authors state, for that to happen successfully it often requires domain knowledge but most importantly current ways to diversify teammates results in “superficial” diversity where trajectories might be different but policies themselves still correlated. Instead, to promote diversity authors propose to use the Best-Response Diversity (BRDiv) metric which directly operates on the returns of the agents. The authors compare their method for diversifying against previous trajectory diversification methods in 3 environments. Results show that the proposed method yields better returns while also resulting in more diverse behaviours.

All reviewers agreed that this is a strong submission with dense yet clear results or a simple method that can be applied across different setups and environments. Reviewers have some issues wrt to clarify and additional discussions added, and authors have incorporated these into their manuscript. The main outstanding points referenced across reviews was, perhaps, the limited and artificial set of environments used here. Reading the reviews, author response and paper, I agree somewhat with the reviewers -- however, there can always be more complex and more environments and LBF is also an environment used for this type of work. Ultimately, all reviewers agreed that this submission provides valuable insights. I agree with them and so I’m going to recommend acceptance.

**Audience:**

Yes

**Claims And Evidence:**

Yes